# Towards designing of a potential new HIV-1 protease inhibitor using QSAR study in combination with Molecular docking and Molecular dynamics simulations

**Mouna Baassi[1], Mohamed Moussaoui[1], Hatim Soufi[1], Sanchaita Rajkhowa[2]\*, Ashwani Sharma[3], Subrata Sinha[2], Said Belaaouad[1]**

**1** Faculty of Sciences Ben M'Sick, Laboratory of Physical Chemistry of Materials, Hassan II University of Casablanca, Casablanca, Morocco, **2** Centre for Biotechnology and Bioinformatics, Dibrugarh University, Dibrugarh, Assam, India, **3** CEO Insight BioSolutions, 35000, Rennes, France

\* s_rajkhowa@dibru.ac.in

**Data Availability Statement:** All relevant data are within the paper.

## Abstract

Human Immunodeficiency Virus type 1 protease (HIV-1 PR) is one of the most challenging targets of antiretroviral therapy used in the treatment of AIDS-infected people. The performance of protease inhibitors (PIs) is limited by the development of protease mutations that can promote resistance to the treatment. The current study was carried out using statistics and bioinformatics tools. A series of thirty-three compounds with known enzymatic inhibitory activities against HIV-1 protease was used in this paper to build a mathematical model relating the structure to the biological activity. These compounds were designed by software; their descriptors were computed using various tools, such as Gaussian, Chem3D, ChemSketch and MarvinSketch. Computational methods generated the best model based on its statistical parameters. The model's applicability domain (AD) was elaborated. Furthermore, one compound has been proposed as efficient against HIV-1 protease with comparable biological activity to the existing ones; this drug candidate was evaluated using ADMET properties and Lipinski's rule. Molecular Docking performed on Wild Type, and Mutant Type HIV-1 proteases allowed the investigation of the interaction types displayed between the proteases and the ligands, Darunavir (DRV) and the new drug (ND). Molecular dynamics simulation was also used in order to investigate the complexes' stability allowing a comparative study on the performance of both ligands (DRV & ND). Our study suggested that the new molecule showed comparable results to that of darunavir and maybe used for further experimental studies. Our study may also be used as pipeline to search and design new potential inhibitors of HIV-1 proteases.

## Introduction

Human Immunodeficiency Virus (HIV) is one of the most challenging viruses in medicine, causing severe complications related to human health [1]. HIV which is responsible for

**Funding:** The authors received no specific funding for this work.

**Competing interests:** The authors have declared that no competing interests exist.

Acquired Immunodeficiency Syndrome (AIDS), still has no cure for more than three decades [2]. This is the main reason why synthesized drugs have been used in combinations to treat HIV infection [3,4]. Highly active antiretroviral therapy (HAART) attacks multiple stages of the HIV viral life cycle and stops the virus from making copies of itself in the body thus leading to a reduction in mortality and morbidity rates of HIV/AIDS [3,5–7].

Antiretroviral therapy plays an essential role in the treatment of HIV/AIDS, but the accelerated evolution of multidrug-resistant (MDR) strains of HIV-1 protease (PR) and poor oral bioavailability and side effects have firmly restricted long-term treatment benefits [8,9].

PIs are supposed to overcome the replication of viruses. However, some residual viral activity endures throughout the therapy process, leading to the development of drug-resistant strains with various mutations that decrease protease affinity for the inhibitors. The mutations are detected not precisely inside the active site where they directly affect the inhibitor binding but also outside the binding site [10–12].

Corresponding to the International AIDS Society, 23 mutations in 16 codons of the protease gene relevant to significant drug resistance to PIs were highlighted using phenotypic resistance assays [13].

Therefore, the design of new HIV-1 PIs has become an obligation. In order to discover new drugs, looking forward to amplifying the inhibitory activity and according to the strategy to defeat drug resistance, a series of 33 compounds were synthesized and evaluated in previous work for their antiretroviral activities [14]. The primary purpose of this study is to develop a Quantitative Structure Activity Relationship (QSAR) model able to relate the structural features (descriptors) to the biological activity of these drug candidates against HIV-1 protease.

The QSAR method is based on computational methods, aiming at relating the activity (y) to the chemical properties (x), y = f(x) [15]. To achieve this, we need a series of compounds with well-known biological activities (y), and for each compound, we compute several descriptors (x) using various software, incorporating the DFT method [16,17].

Once the QSAR model is elaborated and statistically validated, it can be used for the prediction, analysis, and estimation of new elements with convenient activities, minimizing time, effort, and charges [18]. The flow chart mentioned above (Fig 1) covers an overview of the multiple axes elaborated along with this research.

## Material and methods

### Chemical compounds and descriptors

Ten HIV-1 protease inhibitors have been approved by the Food and Drug Administration (FDA), but the emergence of multidrug-resistant (MDR) strains has limited long-term treatment options [19–22]; therefore, the search for new efficient drugs has become a necessity.

Thirty-three new compounds were synthesized and evaluated in a previous study to determine their optimal biological activity [14] (S1 Table in S1 File).

Meanwhile, this work is based on computing various descriptors (Topological, Constitutional, Geometrical, Physicochemical, and Quantum) of the compounds mentioned above using several software packages (Gaussian, Chem3D, ChemSketch, and MarvinSketch).

### Descriptive analysis

The computed descriptors must be analyzed to generate a computational model that relates the biological activity of these compounds to the structure (descriptors).

To do so, we used both methods; the first one is called Principal Component Analysis (PCA), the main purpose of which is to delete correlated descriptors, so we lower the dimension of the data representation area. The second one is a clustering method, called k-means

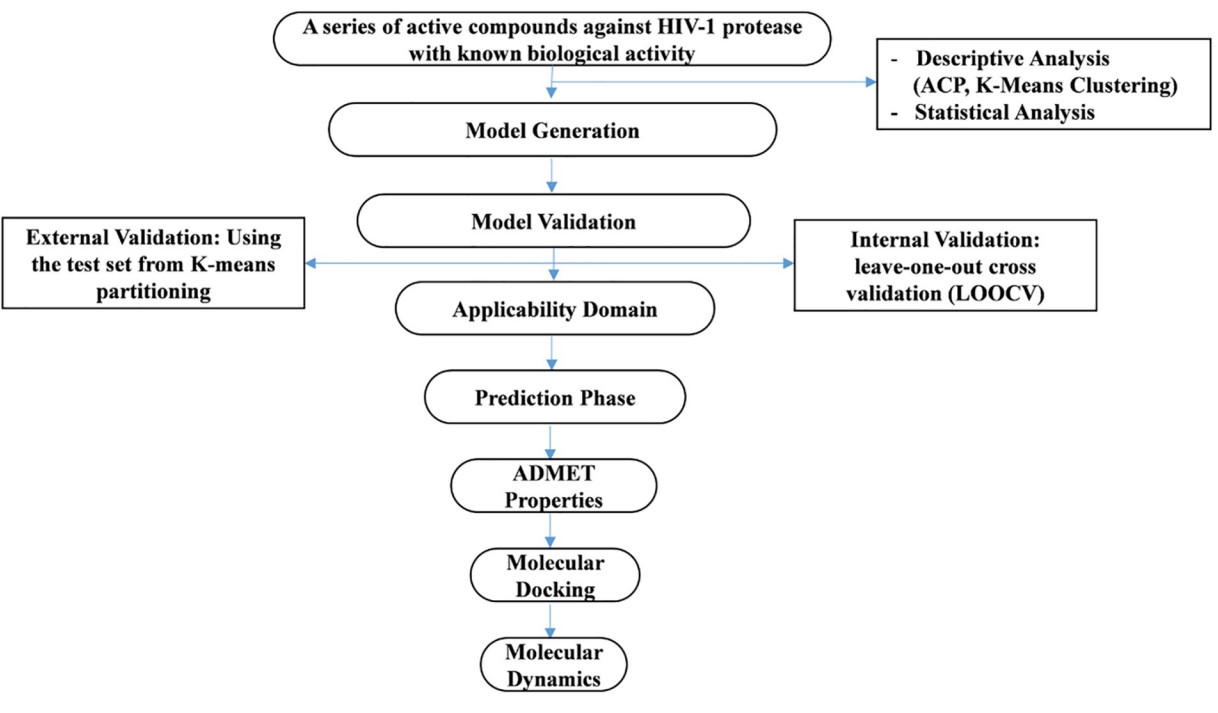

**Fig 1. Flow chart of the current work.**

partitioning, used to split the dataset into training sets for model generation and test sets for validation.

## Statistical analysis

**Multiple linear regression.** Multiple linear regression analysis is a statistical technique based on several analytical independent variables called descriptors to anticipate the outcome of a response variable (biological activity); it is selected to asset a linear model relating the activity (dependent variable) to descriptors having high correlation with the response (activity) [23].

The linear model takes the form that follows:

$$Y = a_0 + \sum_{i=1}^{n} ai \cdot xi$$

Where; $Y$ represents the biological activity (dependent variable), $a_0$ is the intercept of the equation, $x_i$ is the molecular descriptors, and $a_i$ is their coefficients.

**Model generation.** A QSAR model was generated using XLSTAT software after analyzing the data with both methods (PCA and K-means) [24], which after validation, were used to anticipate the activity of brand-new compounds that can be more efficient as HIV-1 protease inhibitors.

In order to assess the physicochemical influence of the substituents (structure/descriptors) on the biological activity, we introduced the dataset along with descriptors corresponding to the 33 compounds listed previously and their biological activities to an MLR analysis.

To choose the first-rate regression performance, we use several coefficients; r, $r^2$, $r^2_{adj}$, MSE and $P_{value}$ [25,26], where r represents the correlation coefficient, $r^2$ is the coefficient of determination, $r^2_{adj}$ is the coefficient adjusted for degrees of freedom, MSE is the mean squared error, and $P_{value}$ is the probability of Fisher statistics.

**Model validation.** The model generated by MLR analysis must be validated to evaluate its significance and ensure its accuracy prediction ability. In order to achieve this, we use internal and external validation.

*Internal validation.* Also called leave-one-out cross-validation (LOOCV), whereby one element is removed from the training set, and the remaining compounds are used to rebuild a model; then it will be returned to the training set, and another compound will be removed, the model generated will be used to predict the activity of the removed one and the cycle is repeated until all compounds have been detached one by one, in the end, a correlation coefficient $Q^2$ is computed [27].

*External validation.* Besides the internal validation, external validation is primordial; the k-means clustering method allowed us to divide the dataset into training and test sets. The second one was employed in this stage. The obtained model will be used to investigate the activities of the test set compounds, and the regression coefficient ($R^2_{cv}$) value will be computed [28].

**Applicability domain.** The model was obtained based on the training set, so it is valid only with compounds with similarities as compared to those included in the training set. Therefore, new molecules must belong to the training domain. A model without an applicability domain can presume the activity of all compounds, regardless of their features, compared to those counted in the aberrant training set. So the AD is a tool to detect compounds outside the applicability domain of the obtained QSAR model and the outliers in the training set [29].

## Molecular docking

Molecular Docking is an important technique used to preview the binding affinities for a vast number of small molecules, with the protease generating several conformations of the ligand-protease complex that will be ranked based on their affinity [30].

The main purpose of molecular docking study is to assess the binding energy as well as the interaction types between the ligands and the protease [31].

## ADMET properties

The Absorption, Distribution, Metabolism, Elimination, and Toxicity (ADMET) properties are crucial for the effectiveness and safety of a therapeutic compound. More than 50% of practical clinical tests are unsuccessful due to the insufficiency in ADMET properties [32]. Therefore, computing ADMET properties using various servers in the drug design field can significantly shorten the probability of drug evolution failure.

These properties can be predicted using many servers, such as pkCSM [33] and SwissADME [34]. The obtained properties contain drug-likeness prediction based on Lipinski's rule. When compounds meet Lipinski's rule with a bioavailability score of 0.55 they will be considered as sufficiently absorbable via oral route [35,36].

## Molecular dynamics simulation

Molecular dynamics simulation is the most incredible tool to predict the properties of new particles and their motion [37]. In this work, we aim to predict the dynamics information between the HIV1-protease and the proposed ligand in order to check the stability results of the docked complex [38]. For the Molecular Dynamics Simulations and MM-PBSA calculations, a similar methodology performed in a previous study was adopted [39].

## Results and discussions

### Chemical compounds

A series of thirty-three compounds (inhibitors with purine base amine-acetamide as P2-ligands) synthesized and evaluated for their biological activities in previous work are the key elements in the current research; their molecular structures are listed in the ST1 Table in S1 File.

### Dependent variable values

The experiment $IC_{50}$, biological activity values, were transformed to the negative logarithm of $IC_{50}$, using the following equation: $pIC_{50}$ = -log ($IC_{50}$). The results are listed in the table below (Table 1).

### Descriptors generation

Several softwares were used to compute various descriptors such as Gaussian, Chem3D, ChemSketch and MarvinSketch, but only some descriptors correlated with the activity were used in minimizing the size of the data representation space. Considering the quantum descriptors, they were investigated using DFT approach performed by Gaussian 09 program package; employing for this purpose the hybrid method B3LYP combining the Becke's three-parameter and the Lee-Yang-Parr exchange-correlation functional, using as well 6-31G (d,p) basis set, performing the optimization of the compounds geometries ultimately while all the other parameters were computed using Chem3D, Chemsketch and MarvinSketch software (S2 Table in S1 File).

### Principle component analysis

Using the Principal Component Analysis, the size of the data representation space was reduced using descriptors that show a correlation coefficient with the activity higher than 0.1

**Table 1. Negative logarithm values of the biological activity concerning the 33 compounds.**

| Number | Compounds | $IC_{50}$ | $pIC_{50}$ | Number | Compounds | $IC_{50}$ | $pIC_{50}$ |
|---|---|---|---|---|---|---|---|
| 1 | 16a | 0.04 | **1.37** | 18 | 18f | 3.76 | -0.58 |
| 2 | 17a | 0.31 | 0.51 | 19 | 16g | 0.15 | 0.82 |
| 3 | 18a | 7.02 | -0.85 | 20 | 17g | 0.57 | 0.24 |
| 4 | 16b | 0.57 | 0.24 | 21 | 18g | 0.64 | 0.19 |
| 5 | 17b | 0.96 | 0.02 | 22 | 16h | 0.18 | 0.74 |
| 6 | 18b | 11.7 | -1.07 | 23 | 17h | 2.60 | -0.41 |
| 7 | 16c | 0.24 | 0.62 | 24 | 18h | 1.51 | -0.18 |
| 8 | 17c | 2.58 | -0.41 | 25 | 16i | 0.46 | 0.34 |
| 9 | 18c | 0.57 | 0.24 | 26 | 17i | 1.73 | -0.24 |
| 10 | 16d | 1.98 | -0.30 | 27 | 18i | 0.19 | 0.72 |
| 11 | 17d | 3.58 | -0.55 | 28 | 16j | 0.07 | **1.17** |
| 12 | 18d | 1.24 | -0.09 | 29 | 17j | 2.43 | -0.39 |
| 13 | 16e | 0.36 | 0.44 | 30 | 18j | 1.81 | -0.26 |
| 14 | 17e | 3.68 | -0.57 | 31 | 16k | 0.08 | **1.10** |
| 15 | 18e | 6.79 | -0.83 | 32 | 17k | 0.79 | 0.10 |
| 16 | 16f | 0.04 | **1.38** | 33 | 18k | 4.73 | -0.67 |
| 17 | 17f | 2.53 | -0.40 | | | | |

**Table 2. Descriptors showing correlation coefficients higher than 0.1 with the activity.**

| Descriptors | r | Descriptors | r |
|---|---|---|---|
| E Gap | 0.24 | Molar Volume | 0.62 |
| Henry's Law Constant | 0.11 | Surface Tension | 0.60 |
| Number of HBond Donors | 0.35 | Density | 0.51 |
| Mol Refractivity | 0.16 | Polarizability | 0.32 |
| Partition Coefficient | 0.54 | Chemaxon HLB | 0.18 |
| LogP | 0.44 | Atom count | 0.38 |
| LogS | 0.10 | Bond count | 0.38 |
| Molecular Topological Index | 0.11 | Dreiding energy | 0.64 |
| Num Rotatable Bonds | 0.36 | van der Waals volume | 0.30 |
| Polar Surface Area | 0.48 | Polar surface area | 0.49 |
| Shape Coefficient | 0.42 | Donor count | 0.35 |
| %C | 0.35 | Donor sites | 0.41 |
| %H | 0.28 | Acceptor count | 0.23 |
| %N | 0.69 | Acceptor sites | 0.11 |
| %O | 0.13 | | |

in absolute value (Table 2), as well as the absence of collinearity between descriptors used to elaborate the model, was inspected by the correlation matrix.

## K-Means Cluster Analysis (k-MCA)

A clustering method, called k-means partitioning, was used to cut the dataset into a training set for model generation and a test set for its validation (Table 3). The data set is divided into five clusters. Five compounds are selected randomly, one from each cluster, to form the test set (16f, 17g, 18d, 16b and 17h), while the remaining compounds will form the training set. The last one is the key element to generate the model, and the first one was used to validate it.

## Multiple linear regression (MLR)

**Model generation.** The model was elaborated using XLSTAT, statistical software, used as add-on for Excel.

*MLR equation*:

pIC50 = $-$ 3$-$0.59*$E_{Gap}$+1.27*HLC$-$ 0.033*PSA$-$ 0.015*DE

*Statistical parameters*:

$R^2$ = 0.66; $R^2_{Adj}$ = 0.60; MSE = 0.18; $P_{value}$<$10^{-4}$; F = 11.23

For the model above, $P_{value}$ is lower than 0.0001, which means that taking the risk of 0.01% by considering the null hypothesis (no effect of the descriptors on the activity) as wrong, therefore, we can assume that the model proposed includes variables with a representative amount

**Table 3. K-means clustering results.**

| Cluster 1 | 16a, 18b, 17c, 16d, 16e, **16f**, 16g, 18h, 16j, 18k |
|---|---|
| Cluster 2 | 17a, 17d, 17e, **17g**, 17j |
| Cluster 3 | 18a, 16c, 18c, **18d**, 18e, 18f, 18g, 18j |
| Cluster 4 | **16b**, 17b, 16h, 18i, 16k |
| Cluster 5 | 17f, **17h**, 16i, 17i, 17k |

**Table 4. Multi-collinearity statistics.**

| Statistic | E Gap | Henry's Law Constant | Polar Surface Area | Dreiding energy |
|---|---|---|---|---|
| Tolerance | 0.31 | 0.28 | 0.15 | 0.62 |
| VIF | 3.23 | 3.50 | 6.69 | 1.61 |

of information. The higher values of $R^2$ and $R^2_{Adj}$ and the lower value of **MSE** show that the proposed model has a higher predictive ability and reliability.

The existence of multi-collinearity among the descriptors was investigated with a parameter called variance inflation factor **(VIF)**, the highest value is less than ten **(VIF = 0.62)** which further confirmed the absence of multi-collinearity problem [40,41]. The table below shows the variance inflation factor values (Table 4).

*Model interpretation*:

In the proposed model, descriptors that are influencing the activity negatively are the Energy Gap ($E_{Gap}$), the Polar Surface Area (PSA) and the Dreiding Energy (DE), while only one parameter has a positive influence on the activity, which is Henry's Law Constant (HLC).

- $E_{Gap}$ displays a negative sign in the model, which means that increasing the activity requires minimizing $E_{Gap}$ value, as well as PSA and DE.

- HLC shows a positive sign in the model, allowing us to conclude that increasing the activity is achieved by increasing HLC.

To sum up, the biological activity is influenced by four variables ($E_{Gap}$, HLC, PSA and DE). To increase the biological activity, $E_{Gap}$ must be decreased, PSA as well as the DE while HLC is increased.

**Internal and external validation.** The model proposed, despite its statistical parameters, must be validated following two steps:

*Internal validation (Y-randomization test)*:

The leave-one-out cross-validation technique obtains the model's cross validation coefficient, the coefficient $Q^2_{LOO}$ obtained is used as a proof of both robustness and predictive capacity of the model [42]. The given model's robustness was confirmed with a cross validation value of 0.53 ($Q^2_{LOO} = 0.53$).

*Y-randomization test*

Y-scrambling is performed on the training set; it is used to confirm that the developed model was not a result of random correlation between the biological activity and the descriptors. In this analysis, the dataset is permuted; the biological activity values were randomly distributed while the descriptors matrix was unchanged, followed by MLR analysis generating new models [43].

For each randomization and subsequent MLR analysis, we obtain a new set of values for $R^2_{Rand}$ and $Q^2_{Rand}$ [44] **(Table ST3)**. If the new QSAR models have lower determination coefficient ($R^2_{Rand}$) and leave one out determination coefficient ($Q_{LOO}^2$) values as well for several trials (100 times in this study), we consider the proposed QSAR model as robust. Moreover, if the $cRp^2$ is greater than 0.5, it will be confirmed that the model is not a result of chance correlation [45,46].

For the current work, the average values of $R_{Rand}$, $R^2_{Rand}$ and $Q^2_{cv\,(Rand)}$ are 0.35, 0.14 and -0.29 respectively, the $cRp^2$ value equals 0.60 which is higher than 0.5 (S3 Table in S1 File), and all the new QSAR models are showing significantly lower $R^2_{Rand}$ and $Q^2_{cv\,(Rand)}$ values for the 100 trials. Therefor Y-randomization analysis results are showing that there is no

**Table 5. Descriptors' computed values and predicted activities as well of the test set compounds using the MLR model generated.**

| Test set | E$_{Gap}$ | HLC | PSA | DE | pIC50 (Obs) | pIC50 (Pred) |
|---|---|---|---|---|---|---|
| 4 (16b) | 4.85 | 12.31 | 162.28 | 261.65 | 0.24 | 0.57 |
| 12 (18d) | 5.11 | 12.18 | 179.07 | 290.34 | -0.09 | -0.73 |
| 16 (16f) | 5.19 | 11.61 | 136.26 | 230.04 | 1.38 | 0.80 |
| 20 (17g) | 3.86 | 11.61 | 178.84 | 248.79 | 0.24 | -0.08 |
| 23 (17h) | 3.40 | 11.80 | 188.07 | 279.42 | -0.42 | -0.34 |

random correlation between the activity and the descriptors affecting significantly the response and the developed QSAR model is robust.

**External validation**:

The model then must be externally validated using the test set mentioned above, in this stage, the model proposed must conclude the activities of the test set compounds in arrangement with the experimental values (Table 5), graphically presented in the figure bellow (Fig 2). The predictive ability was confirmed with a test coefficient value of 0.64 ($R^2_{Test}$ = 0.64).

## Applicability domain (AD)

The standardized residuals and the leverage were both jointed to illustrate the applicability domain. The Williams plot for the QSAR model is illuminated in figure below (Fig 3). The warning leverage (h*) was found to be 0.45 for the developed QSAR model. Based on the leverages, all compounds were found to be inside the defined AD.

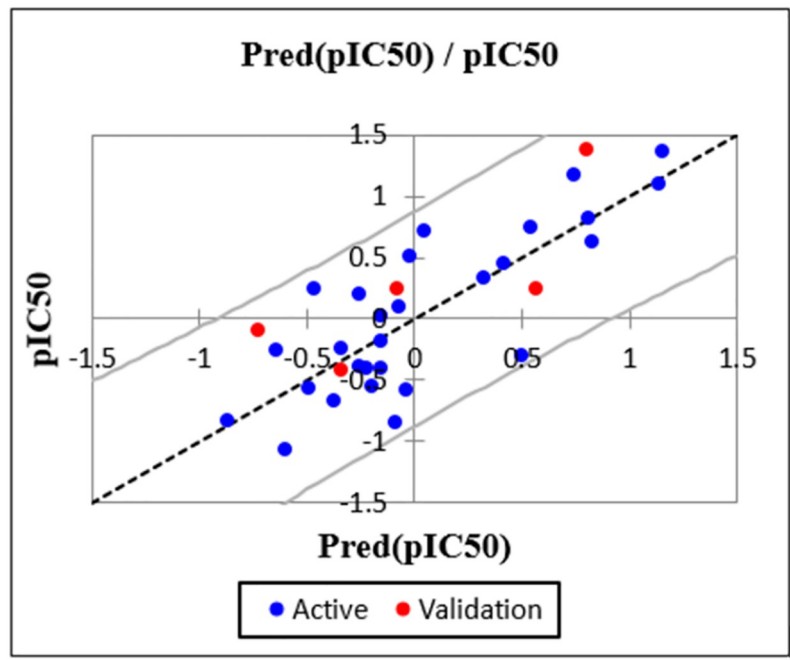

**Fig 2. Correlation of observed and predicted activities (training set in blue and test set in red).**

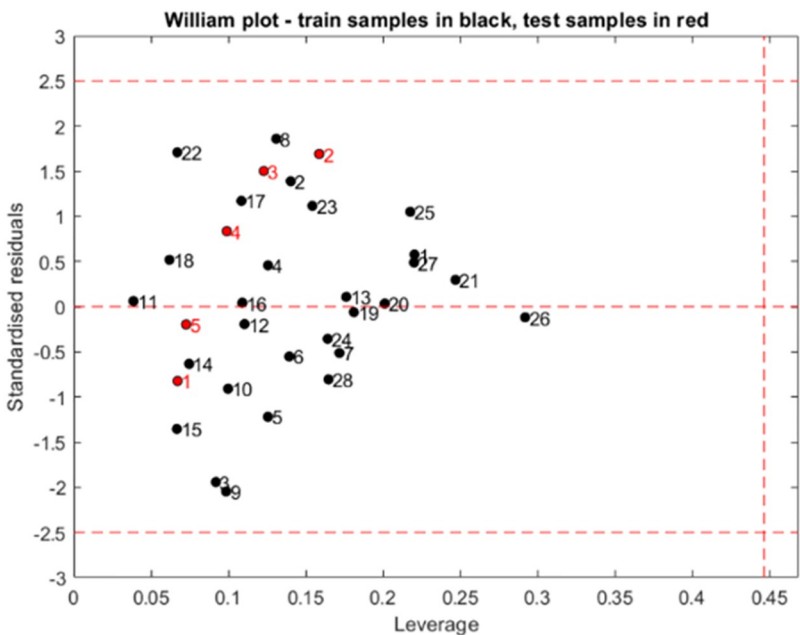

**Fig 3. Williams plot of standardized residual versus leverage for the MLR model (with: h\* = 0.45 and residual limits = ± 2.5); training samples are designed in black color and test samples in red color.**

## New drugs elaboration

In order to suggest new efficient compounds, we must select from the series of compounds used in the present work, those with the highest values of pIC50 (1.37, 1.38, 1.17, 1.10) corresponding to (16a, 16f, 16j and 16k) respectively. These particles will be the object of structural modification in order to design new molecules; their descriptors' values are determined using the same tools as well as $pIC_{50}$ values predicted by MLR model proposed. Furthermore, 24 compounds candidates were designed and their parameters were computed. The leverage values (hi) were computed using Matlab software with the following equation: $\mathbf{hi = x_i^T (X^TX)^{-1}x_i}$ $(\mathbf{i = 1, 2 \ldots n})$ (S4 Table in S1 File).

With: $\mathbf{x_i}$ represents the proposed compounds descriptors' matrix, $\mathbf{X}$ represents the test set descriptors' matrix and $\mathbf{X^T}$ represents the transpose of the test set descriptors' matrix.

Among the 24 compounds, only one compound ($\mathbf{16^{th}}$) has a leverage value ($\mathbf{h_i = 0.43}$) lower than h\* (h\* = 0.45) and a biological activity higher than the known ones ($\mathbf{pIC_{50} = 1.58}$) (Fig 4).

## ADMET properties

In the one hand, regarding Lipinski's rule, the drug-likeness of the proposed compound was verified with only one violation (MW>500) (Table 6), which means that the proposed compound is considered as sufficiently absorbable via oral route with a bioavailability score of 0.55, in the other hand, ADMET properties predictions for the selected compound were performed using SwissADME and pkCSM web servers.

The pharmacokinetic parameters (ADMET) (absorption, distribution, metabolism, excretion, and toxicity) related to the brand-new drug are computed using pkCSM.

The **absorption** of the drug is primarily based on the factors that comply with; water-solubility, membrane permeability (Caco-2), intestinal absorption (human), skin permeability, p-

**Fig 4. Chemical structure of the new proposed drug (C₂₇H₃₂N₆O₄S).**

glycoprotein. The drug **distribution** properties are expected from the data of volume distribution (VDss), the fraction of unbound drug, the blood-brain barrier (BBB), and central nervous system (CNS) permeability. For the **biotransformation evaluation**, participants of the cytochrome P450 (CYP) superfamily are selected (CYP 2D6, CYP 3A4, CYP 1A2, CYP 2C19, CYP 2C9, CYP 2D6 & CYP 3A4), while the **excretion** of compounds involves the total clearance of xenobiotics and renal clearance via organic cation transporter 2 (OCT 2). The **toxicity** of compounds is investigated using AMES toxicity; maximum tolerated dose, the human Ether-a-go-go Related Gene (hERG) potassium channel inhibition, oral rat acute toxicity, oral rat chronic toxicity, skin sensitization, T.Pyriformis toxicity and Minnow toxicity. Just a few of the important factors are mentioned in the present study, notably:

### Water solubility

For the oral administrative drugs discovery, water solubility prediction is highly required. The decimal logarithm of the molar solubility in water is -3.224 (log mol/L). Considering what follows (Insoluble < -10 < poorly soluble < -6 < Moderately < -4 < soluble <

**Table 6. Physicochemical properties.**

| Formula | $C_{27}H_{32}N_6O_4S$ | Num. rotatable bonds | 13 |
|---|---|---|---|
| Molecular weight | 536.65 g/mol | Num. H-bond acceptors | 8 |
| Log P | 2.26 | Num. H-bond donors | 2 |
| Num. heavy atoms | 38 | Molar Refractivity | 144.01 |
| Num. arom. heavy atoms | 21 | TPSA | 138.69 Å$^2$ |

-2 < very soluble < 0 < highly soluble) [47], the compound has a good solubility in water, therefore the development and the production as well of oral solid dosage is possible.

### Caco2 permeability

If the predicted Papp log value is higher than 0.90 $10^{-6}$ cm/s [47], the compound is considered to have high Caco-2 permeability, for the drug candidate, it has for value 1.098 $10^{-6}$, so we can say it has a high permeability in Caco-2.

### Intestinal absorption (human)

The quantity absorbed of the drug candidate by the intestinal system is one of the major factors for oral bioavailability [48]. For the proposed compound, the intestinal absorption (human, % absorbed) seems to be 74.616%.

### BBB permeability

The BBB permeability of the drug candidate has a value of -1.118 log BBB. According to the research [33], the compound is adept to cross the blood–brain barrier, if the Log BB value is higher than 0.3 and it can't cross adequately the blood–brain barrier if the log BB value is lower than -1. Therefore, the drug candidate won't be able to cross the blood-brain barrier.

### CYP2D6 substrate

Drug that inhibit or compete for CYP2D6 can conduct clinical problems; this isoenzyme is highly polymorphic and is responsible for metabolizing relatively 25% of known pharmaceuticals [49]. In the current study, the drug candidate is not inhibitor of CYP2D6 enzymes.

### Total Clearance

The compound has a total clearance of 0.288 log ml/min/kg, therefore, it could be excreted quickly [47].

### AMES toxicity

The compound is AMES negative and test suggests that the compound could be not mutagenic [47].

### hERG inhibitor

Drugs that block these HERG K+ channels are likely to cause cardiac toxicity [50].
The safe range for an ideal drug should be -5 or higher, if the value is below this level, it is predicted to cause cardiac toxicity [47].

### Oral Rat Acute Toxicity (LD50)

The proposed compound is dangerous only at huge doses regarding its high LD50 value (2,259 mol/kg) [50].

## Molecular docking

Molecular docking study was carried out with the aim of predicting the best conformation of the HIV-1 protease of both types (mutant and wild), on the one hand; combined to the proposed compound as a new efficient drug candidate (ND), on the other hand; combined to an FDA approved drug called Darunavir (DRV). We selected both types of the HIV-1 protease (WT and MT) as receptors. The structures of the wild type (WT) as well as the mutant type (MT) proteases were downloaded from Protein Data Bank (PDB), their PDP codes are respectively: (**4LL3**-*Structure of wild-type HIV-1 protease in complex with Darunavir*) (Fig 5) and (**3TTP**-*Structure of multiresistant HIV-1 protease in complex with Darunavir*) (Fig 6). Their original ligands were eliminated using Discovery Studio, polar hydrogens were added and the

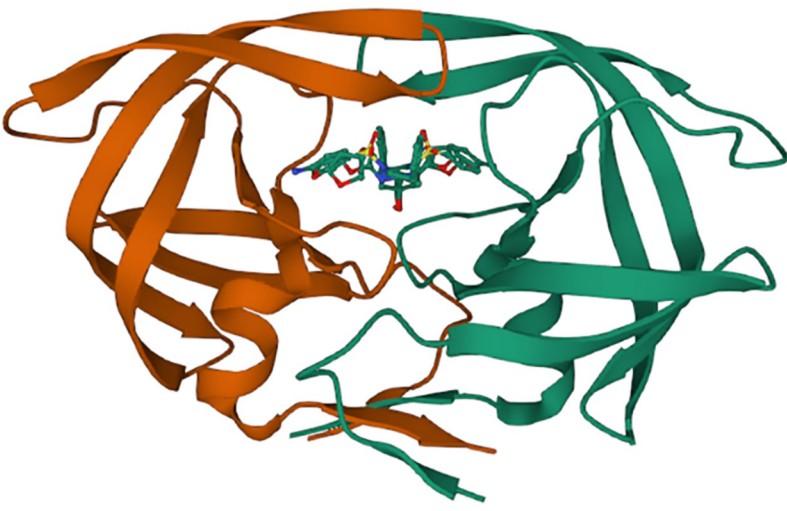

**Fig 5. 4LL3-Structure downloaded from PDB.**

proteins were saved in PDB format, and then saved in PDBQT format using Autodock MGL Tools. The ligand proposed as a new efficient drug was earlier designed and optimized using Gaussian, then saved in PDBQT format by Autodock MGL tools (Fig 7); in addition, DRV was taken from the crystal structures downloaded from Protein Data Bank (Fig 8).

Command prompt and Vina folder were used in order to run the Docking. Different conformations of the ligand binding modes for both types were obtained with their respective binding energies (kcal/mol) after the accomplishment of the docking runs; the best pose is the one with the lowest affinity value.

The best-ranked poses based on their binding affinities are selected for farther analysis; figures (Figs 9–12) represent the 2D-binding interactions in the active site of the proteases; wild type and mutant type with Darunavir and the new drug. Figures (Figs 13–16) disclose the 3D-interactions for the same compounds (WT-ND, MT-ND, WT-DRV & MT-DRV).

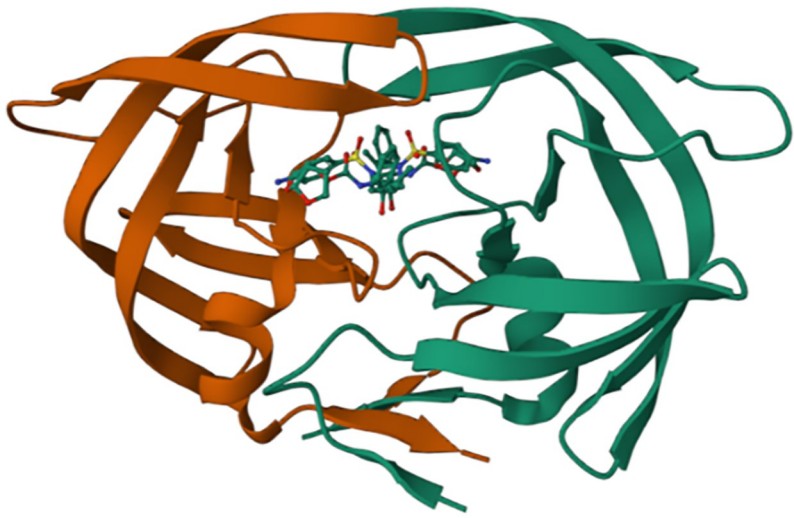

**Fig 6. 3TTP-Structure downloaded from PDB.**

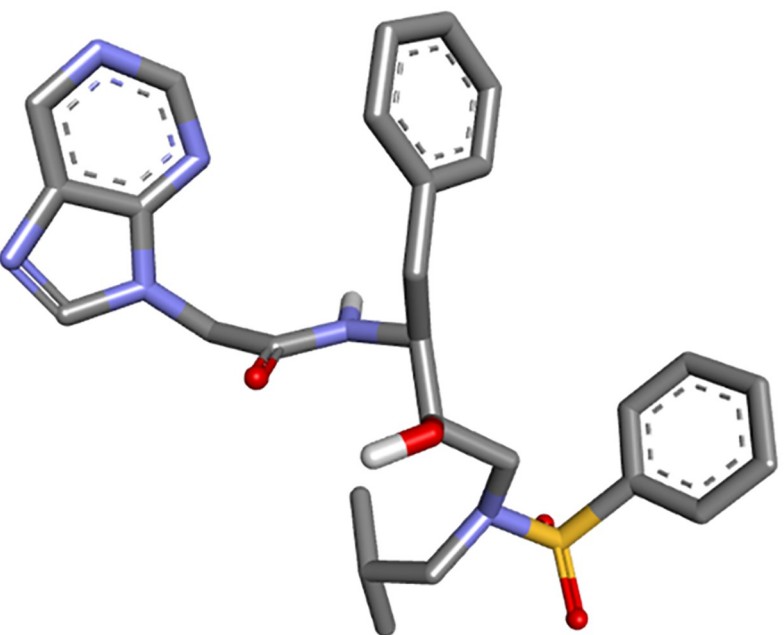

**Fig 7. New drug candidate optimized using Gaussian.**

The interactions between the ligands (ND & DRV) and the proteases were visualized using Discovery Studio (Table 7). Active residues interacting with the ligands (ND & DRV) are also disclosed (S5 Table in S1 File). Moreover, atoms from ligands and residues interacting with each other to form hydrogen bonds are mentioned (S8 Table in S1 File).

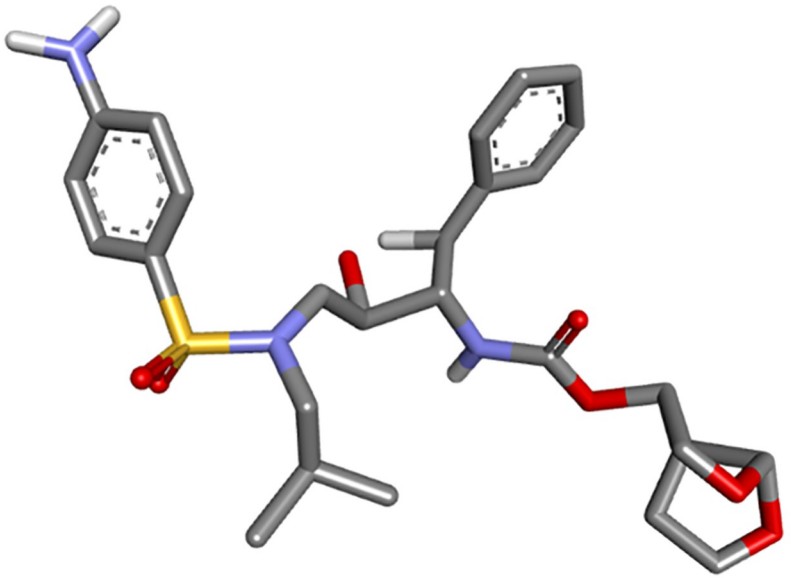

**Fig 8. 3D structure of DRV.**

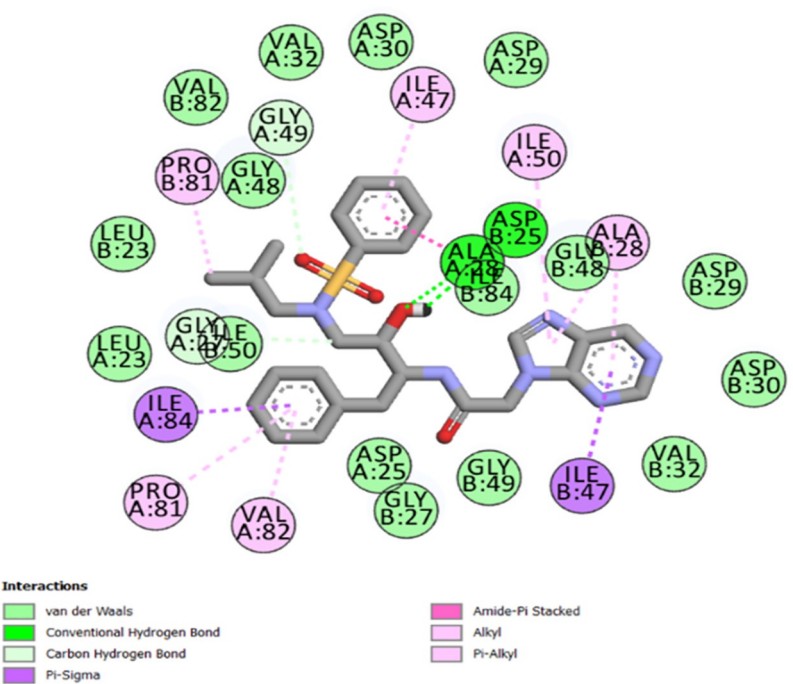

**Fig 9. 2D-binding interactions in the active site of the wild type protease (WT-ND).**

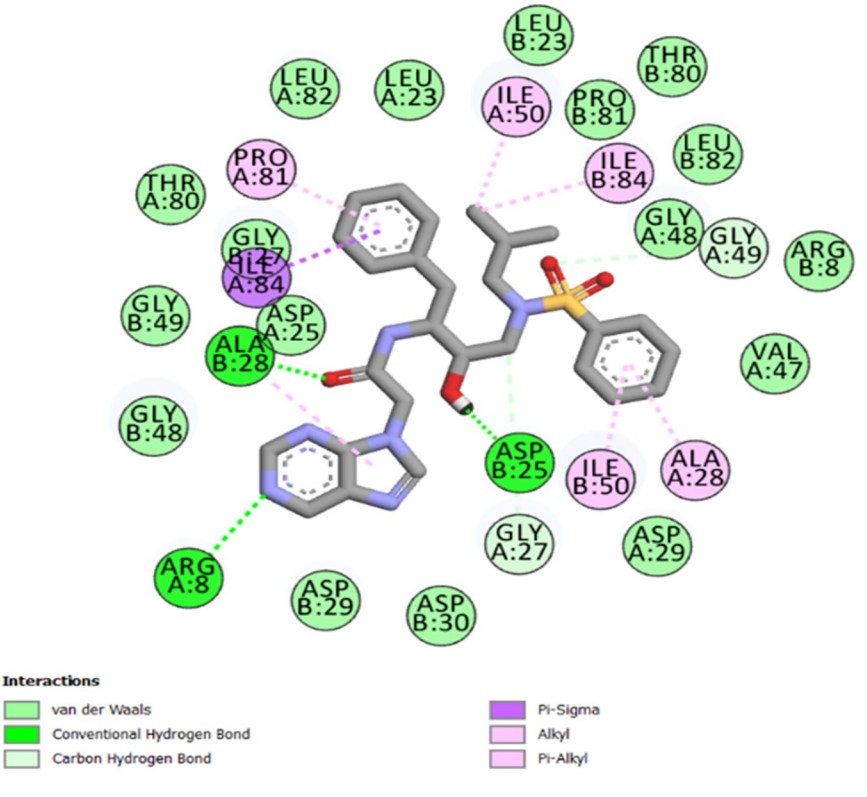

**Fig 10. 2D-binding interactions in the active site of the mutant type protease (MT-ND).**

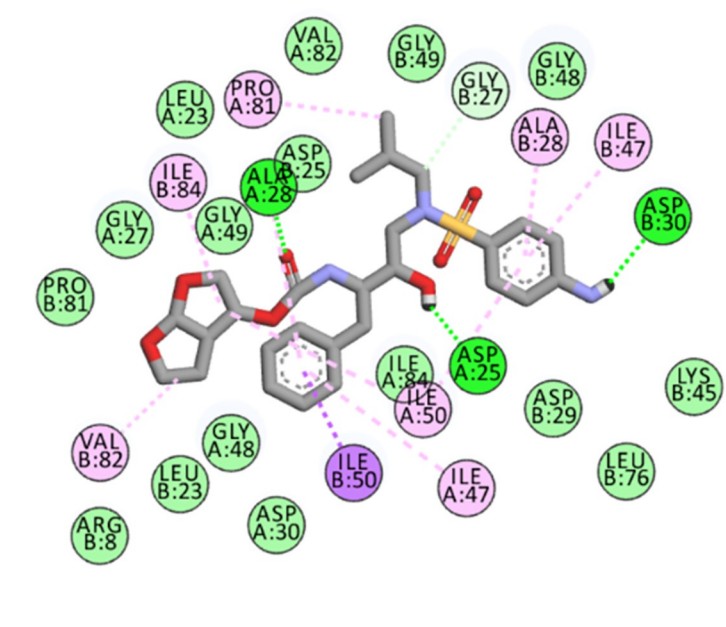

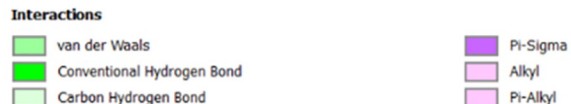

**Fig 11. 2D-binding interactions in the active site of the wild type protease (WT-Darunavir).**

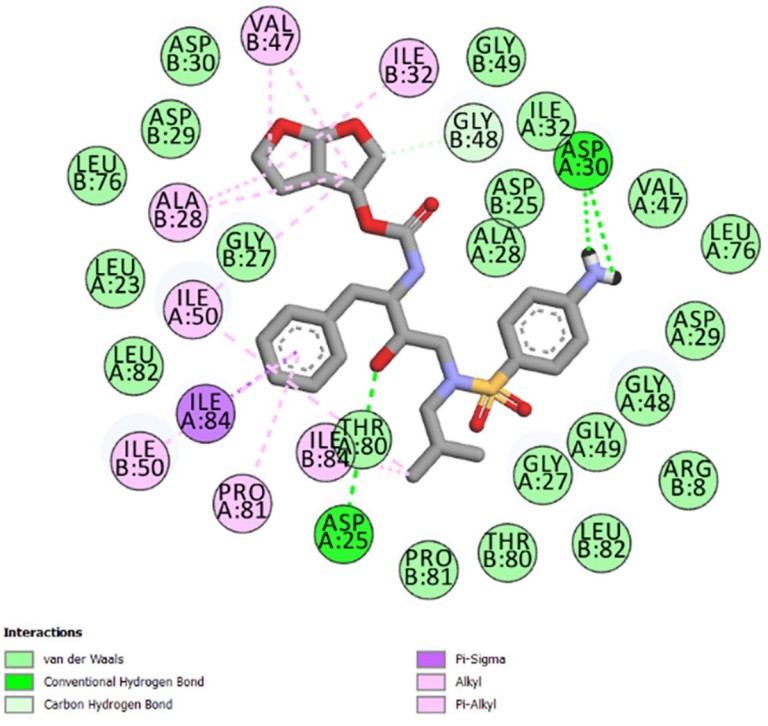

**Fig 12. 2D-binding interactions in the active site of the mutant type protease (MT-Darunavir).**

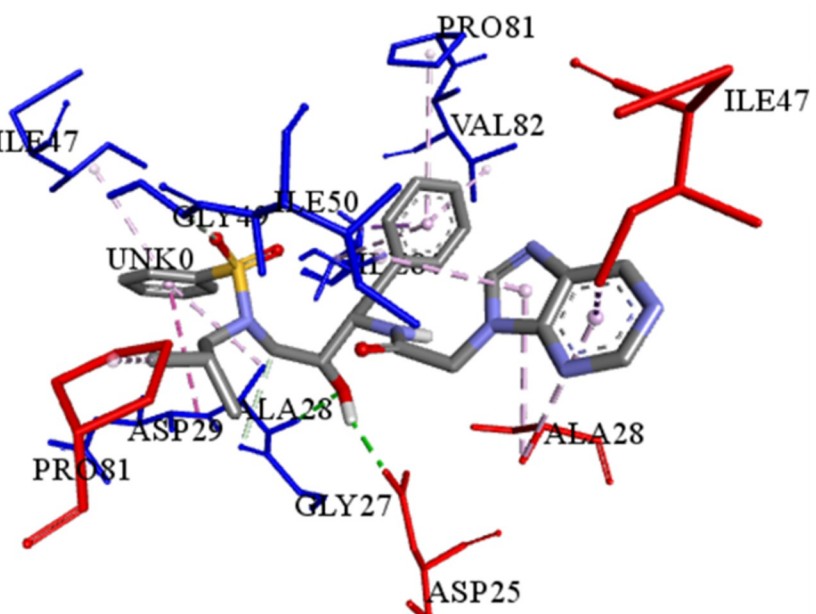

**Fig 13. 3D-binding interactions in the active site of the wild type protease (WT-ND).**

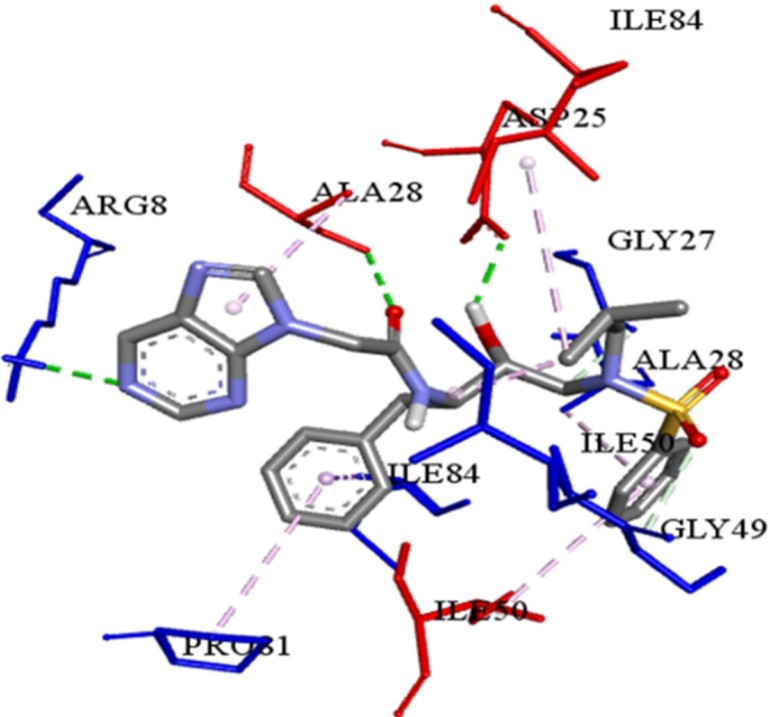

**Fig 14. 3D-binding interactions in the active site of the mutant type protease (MT-ND).**

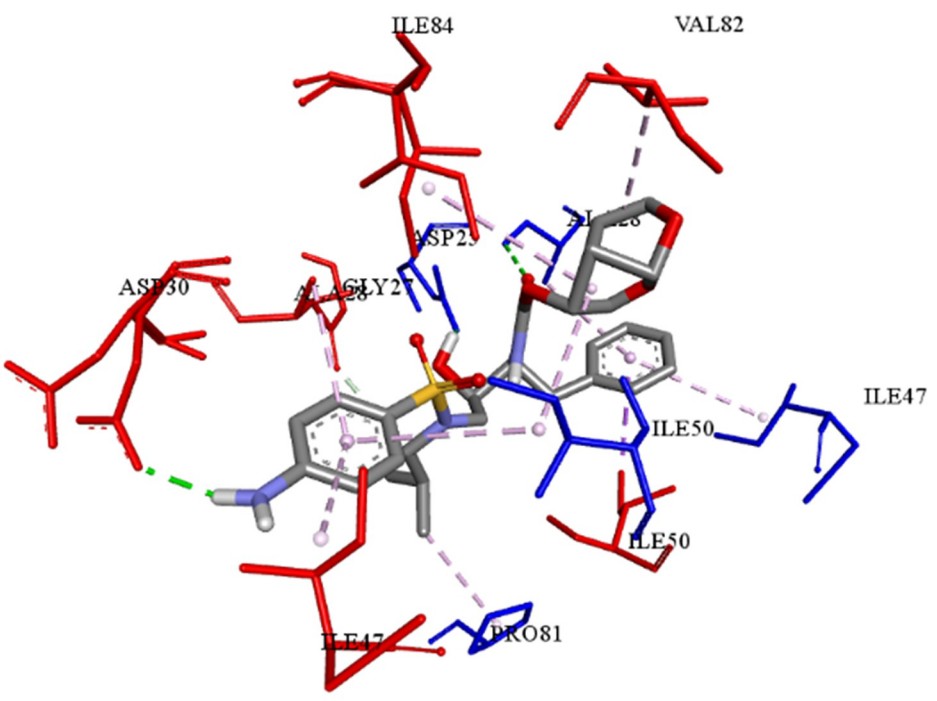

**Fig 15. 3D structure of complex compound with WT protease (WT-DRV).**

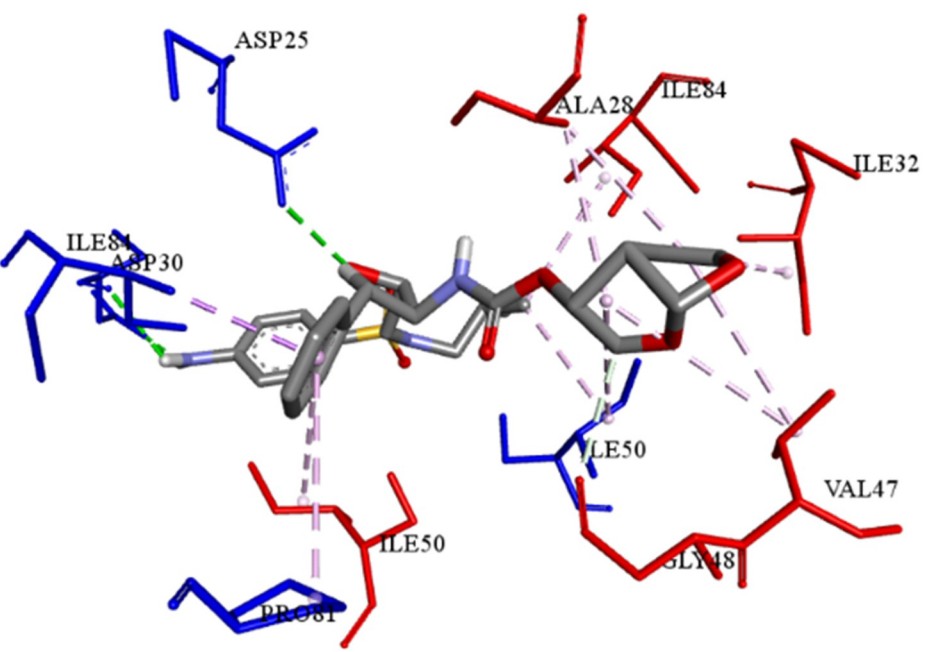

**Fig 16. 3D structure of complex compound with MT protease (MT-DRV).**

**Table 7. Complex compounds' affinity values and the number of different interaction types.**

| Protein | Ligand | Nbre of Hydrogen bonds | Nbre of hydrophobic interactions | VDW | Binding affinity |
|---------|--------|------------------------|----------------------------------|-----|------------------|
| WT | DRV | 4 | 10 | 16 | -9.6 |
| | ND | 4 | 12 | 15 | -10.2 |
| MT | DRV | 4 | 11 | 21 | -9.9 |
| | ND | 5 | 7 | 17 | -10.4 |

Based on the Molecular Docking analysis; results lead us to conclude that the complex compounds (WT-ND & MT-ND) with binding affinity values of -10.2 kcal/mol & -10.4 kcal/mol respectively, display a higher stability as compared to (WT-DRV & MT-DRV).

## Molecular dynamics simulation

To evaluate the native proteins' stability (WT & MT), as well as the docked compounds' (WT-DRV, WT-ND, MT-DRV & MT-ND), a computational process is carried out through the Molecular Dynamics simulation (MD) study, allowing structural analysis at the atomic level, aiming at investigating the motion of the four complex compounds and the native proteins.

Therefore, MD simulations were administered in nine plots, with 100ns for each, using the best poses generated based on the docking study performed previously, the compounds were carried out in water simulations separately. Further, the stability analysis was performed through several techniques, namely: Root Mean Square Deviation (RMSD), Root Mean Square fluctuation (RMSF) and the Radius of Gyration (Rg).

**Root Means Square Deviation (RMSD).** RMSD stands for Root Means Square Deviation, it is a numerical measurement, it estimates the approximate distance between a band of atoms, mainly, backbone atoms of a protein plotted against time. The Root Means Square Deviation value is typically a measure of how much the protein's structure has been modified over time in comparison to the starting point. Further, if the RMSD of the protein presents considerable fluctuations, then no equilibrium is reached, therefore, more simulation time is required for better results.

As the RMSD plots display (Fig 17), the native proteins (WT and MT) do not show any promising stability within the simulation time especially for the wild type protease. Regarding the RMSD plots (Fig 17 (WT)) for the two complexes (WT-DRV and WT-ND), it is highly clear that these compounds are showing lower fluctuations than the native protein (WT) within the simulation time. As for the complexes (MT-ND and MT-DRV), they're showing as well lower fluctuations as compared to the native protein (MT) within the simulation time (Fig 17 (MT)). However, WT-ND and MT-ND complexes are showing promising results comparable to those of Darunavir in terms of fluctuations.

**Root Means Square Fluctuation (RMSF).** The Root Mean Square fluctuation (RMSF) measures the approximate deviation of a particle over time from a reference position at a specific temperature and pressure. The RMSF analysis illuminates the fluctuations of residues during the MD simulation time.

Considering the graphics, for the wild type and the mutant type proteases for both chains (Fig 18), A & B chains are displaying slightly similar fluctuations in some regions, and highly non-similar fluctuations in the other regions, leading us to conclude that for all complexes (WT-DRV, WT-ND, MT-DRV & MT-ND) regardless the chain, the new drug and

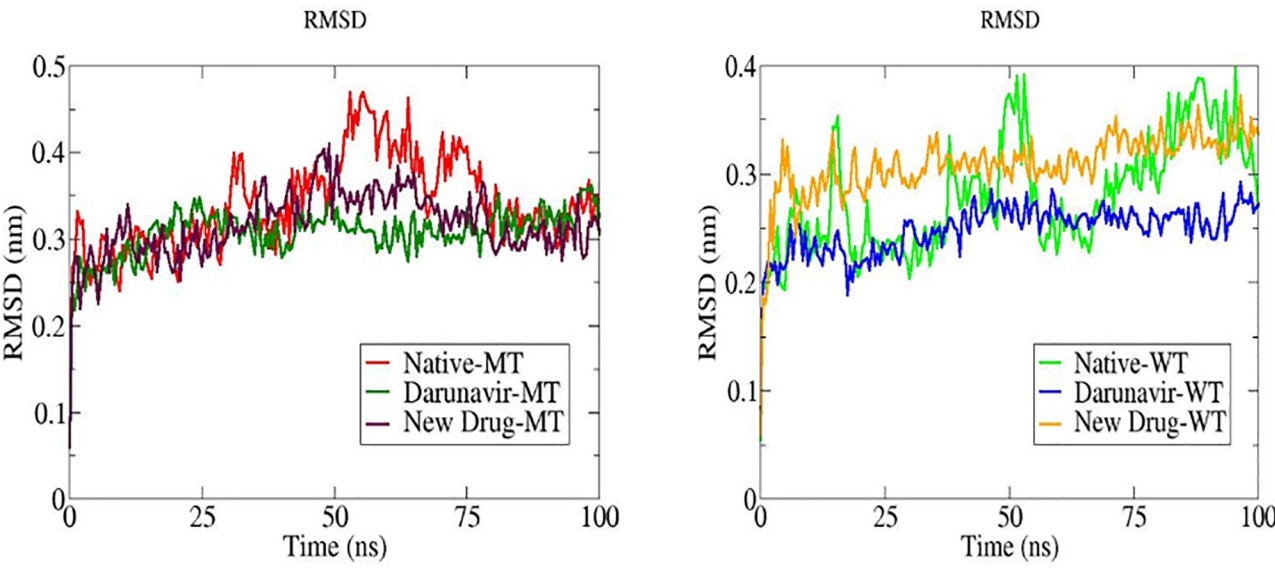

**Fig 17. The root means square deviation (RMSD) plots of MT and WT proteases virgin as with ligands (ND and DRV) during 100ns of molecular dynamics simulation.**

Darunavir are significantly influencing the fluctuations of the proteins' residues in most regions.

**Radius of gyration (Rg).**  The radius of gyration is an interesting parameter as well to investigate the motion of a protein as well as its stability; it describes the compactness of the protein during the simulation time.

For the Mutant Type protease (Fig 19 (MT)), the radius of gyration of the complex compound MT-ND is higher in value as compared to the MT native protein and the complex compound in presence of DRV, causing eventually higher flexibility of the compound MT-ND. For the Wild Type protease (Fig 19 (WT)), the plots show that the complex compound WT-ND reveals more compactness with lower radius of gyration values as compared to the complex compound WT-DRV and the WT native protein within the simulation time, inducing less flexibility, which means higher potential of stability for the complex WT-ND.

**Hydrogen bonds.**  Hydrogen bonds are primordial in drug specificity and stability, so the determination of H-bond number in complex compounds is essential to check its contribution to the overall stability of each system and further conduct a comparative study including all complex compounds in question.

The figure (Fig 20) shows that during the MD simulation period (100ns), the complex MT-ND's graph is showing up to seven hydrogen bonds by the end of the simulation time, while the MT-DRV complex compound's graph is showing a few hydrogen bonds during the first 40ns as compared to MT-ND, then significantly increasing at 60ns displaying ten hydrogen bonds then decreasing to seven by the end of the simulation time (Fig 20 (MT)). In contrast, for the complex compound WT-DRV, the number of hydrogen bonds is consistently decreasing from 8 to 5 while the compound WT-ND displays up to five hydrogen bonds with no significant decrease compared to the WT-DRV compound during the simulation time (Fig 20 (WT)).

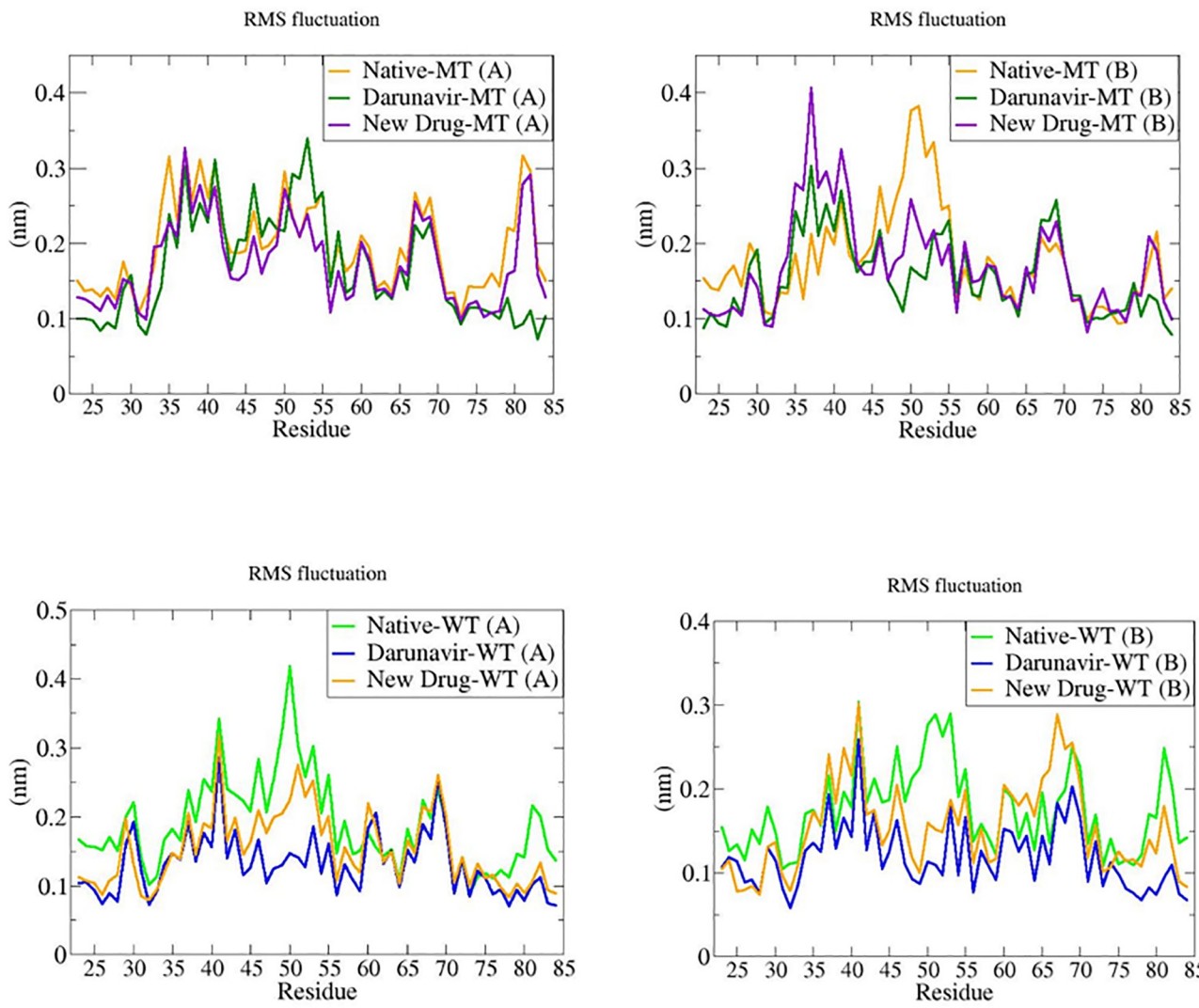

**Fig 18. The root means square fluctuation (RMSF) plots of MT (chain (A) and chain (B)) and WT (chain (A) and chain (B)) proteases without and with ligands (ND and DRV) during the period of simulation.**

We can conclude that whether the wild type or the mutant type proteases, when docked to the new drug, the number of hydrogen bonds is likely to be the same with no significant change as compared to the complex compounds with Darunavir that shows a decreasing number of hydrogen bonds during the simulation time.

**Hydrophobic interactions.** Hydrophobic interactions are non-bonded interactions between the protein and the ligand, which play a major role in the stability of complexes.

As shown below, considering the wild type protease (Fig 21 (WT), both complexes WT-DRV and WT-ND show highly similar numbers of hydrophobic interactions during the simulation time. In contrast, for the mutant type protease (Fig 21 (MT), the complexes MT-DRV and MT-ND, the number of hydrophobic interactions for the complex compound MT-DRV is significantly higher than the number of hydrophobic interactions for the complex compound with the new drug MT-ND.

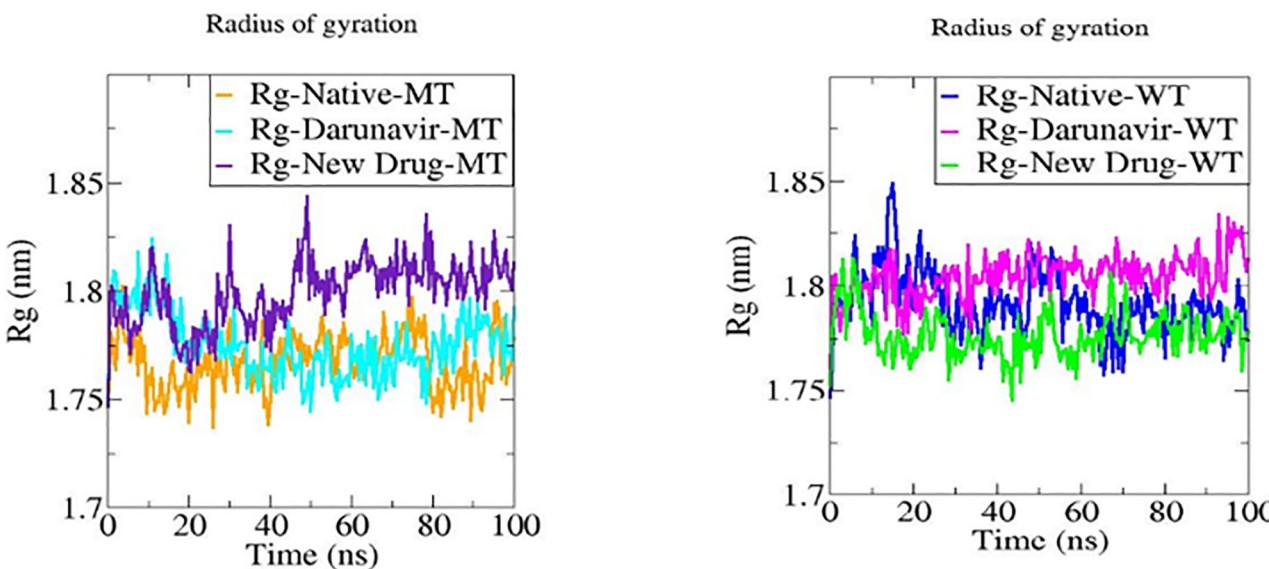

**Fig 19. Graphs representing the Radius of gyration (Rg) values for MT and WT proteases without as with ligands (ND and DRV) during the period of simulation.**

We can conclude that for the wild-type protease, the new drug significantly competes with Darunavir, displaying similar numbers of hydrophobic interactions at every 20 ns of the simulation time. However, Darunavir is showing highly promising results for the mutant-type protease compared to the new drug in terms of hydrophobic interactions.

**Solvent Accessible Surface Area (SASA).** The accessible surface area (ASA) or solvent-accessible surface area (SASA) is the surface area of a biomolecule that is accessible to a solvent.

Based on the graphics (Fig 22), the new drug, when combined to the wild type protease, is showing promising results regarding the significant decrease of the ASA values since 40ns to the end of the simulation time (Fig 22 (WT)), but for the mutant type, the ASA values are not promising on the ground that the graphics are displaying increasing values starting from 60ns of the simulation time (Fig 22 (MT)).

We can conclude that the new drug is comparable to Darunavir during the last 30ns of the simulation time for the wild type protease while no possible competition is investigated for the mutant type on the ground that the graphic is showing significant ASA values for the complex MT-ND as compared the MT-DRV mainly during the last 40ns of the simulation time.

**Binding free energy calculation.** Molecular dynamics simulations were used to calculate binding free energy using the MM-PBSA method. Snapshots were extracted at every 1 ns of stable intervals from 70–100 ns MD trajectory. The binding free energy and its corresponding component obtained from the MM-PBSA calculations are listed (Table 8).

The results indicate that for both wild and mutant type protease, Darunavir is showing a binding affinity of **-173.323 kJ/mol** and **-190.868 kJ/mol**, respectively, which is slightly higher than the New Drug (**-170.903 kJ/mol** and **-187.521 kJ/mol**, respectively).

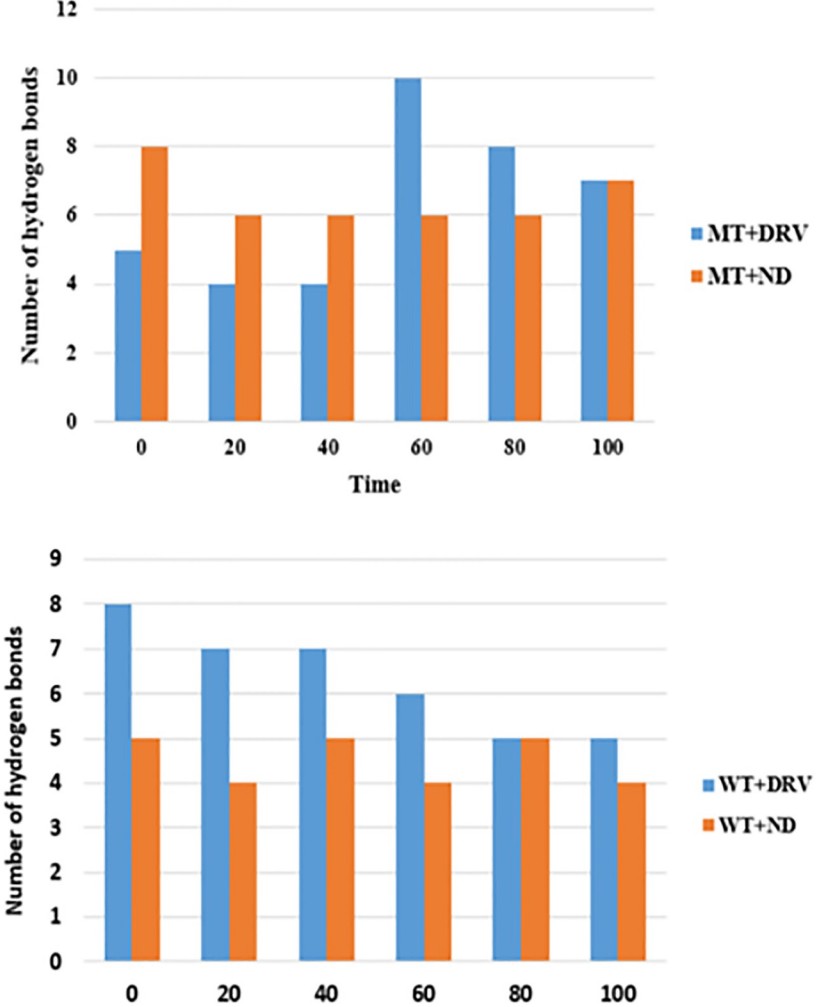

**Fig 20. Graphs showing the number of hydrogen bonds (at every 20 ns) along with the simulation time for complex compounds containing MT and WT proteases.**

van der Waals, Electrostatic and SASA energy played a crucial role in binding energy and complex stability. In contrast, polar solvation energy has an opposite effect causing binding energy to depend on its unfavorable positive value. Among different energy terms, the contribution of van der Waals energy towards total binding energy is superior.

Compilation of the data demonstrated that although the binding of Darunavir to both wild and mutant HIV protease is better, the binding of the new drug is comparable to that of Darunavir in both wild and wild mutant type. This is illustrated by the different analyses that have been used so far. Thus, the new drug may also be considered a potential inhibitor against multi-drug resistant HIV and may be tested experimentally.

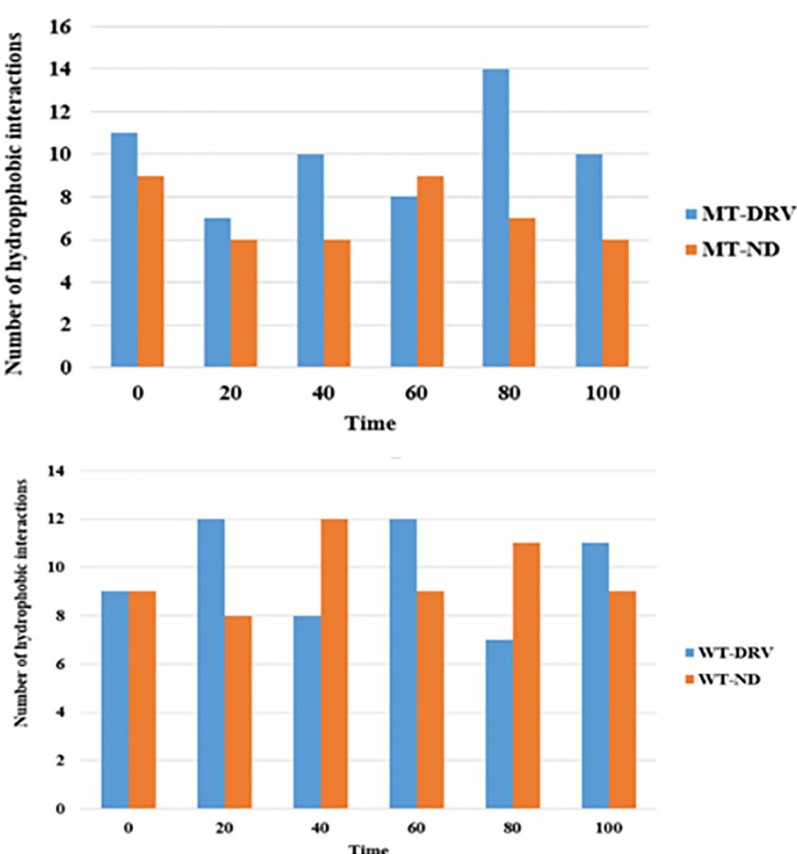

**Fig 21. Graphs showing the number of hydrophobic interactions (at every 20 ns) along with the simulation time for complex compounds containing MT and WT proteases.**

## Conclusion

Various softwares have been used in this study in order to generate a reliable model relating the biological activity of new HIV-1 protease inhibitors to their physicochemical parameters. The generated model showed a high predictability efficiency regarding its statistical parameters. The applicability domain was also generated to frame the workspace (only compounds with features with greater similarity to those included in the training set can be used). Regarding the proposed model, the biological activity of the new HIV-1 protease inhibitors can be increased by increasing the three variables' values; the Energy Gap ($E_{Gap}$); the Polar Surface Area (PSA) and the Dreiding Energy (DE) (positively related to the activity), and decreasing the Henry's Law Constant value (negatively related to the activity). A new drug was proposed based on the model generated with a biological activity higher than the known drug compounds' activities. Afterwards, the molecular docking study was performed on the wild-type and the mutant-type HIV-1 proteases to predict the best conformation displayed by two ligands, the New Drug and Darunavir as an approved FDA drug. Moreover, molecular dynamics simulation was performed to study the stability of the complexes (WT-DRV, WT-ND, MT-DRV & MT-ND); results disclosed some interesting results related to the new drug, therefore, the new drug may be considered as a potential inhibitor against multi-drug-resistant (MDR) strains of HIV-1 protease (PR) and may be tested experimentally.

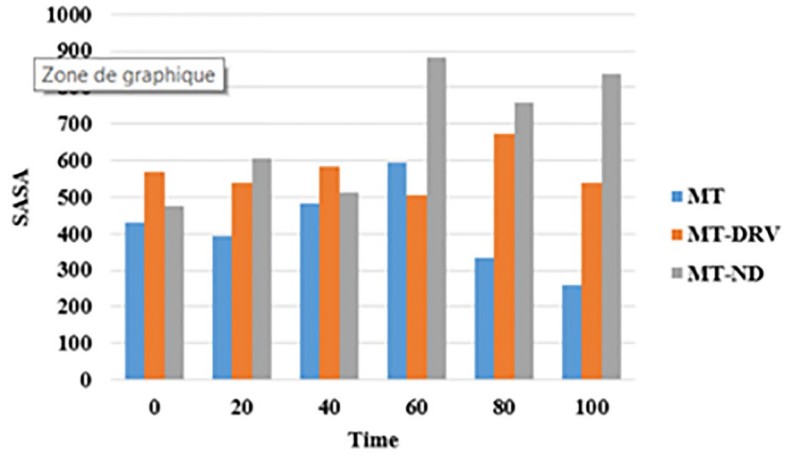

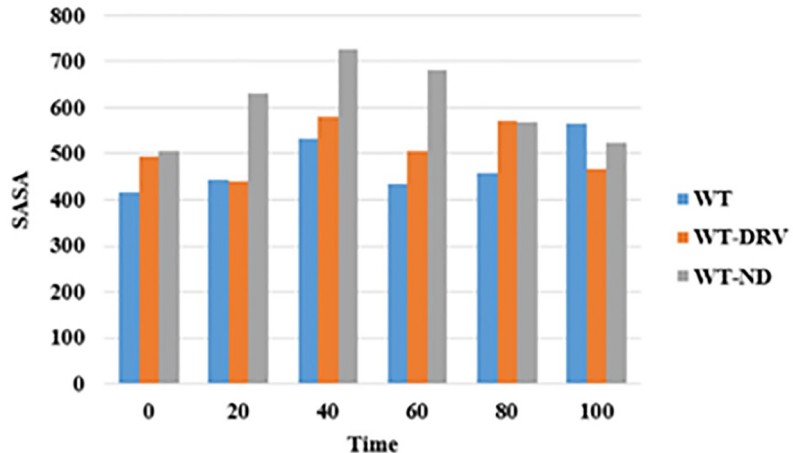

**Fig 22. Time evolution (at every 20 ns) of solvent accessible surface area (SASA) or of the wild type and the mutant type proteases along with DRV and the ND.**

**Table 8. Average MM-PBSA free energies for proteins (WT and MT) and ligands (DRV and ND).**

| Energy | WT-Darunavir (kJ/mol) | WT- New Drug (kJ/mol) | MT- Darunavir (kJ/mol) | MT- New Drug (kJ/mol) |
|---|---|---|---|---|
| **van der Waal** | -225.893 +/- 16.265 | -219.745 +/- 17.427 | -250.856 +/- 22.852 | -248.746 +/- 26.755 |
| **Electrostatic** | -119.917 +/- 24.965 | -75.793 +/- 20.319 | -110.692 +/- 39.922 | -98.412 +/- 26.172 |
| **SASA Energy** | -21.913 +/- 1.188 | -22.853 +/- 1.809 | -22.152 +/- 1.964 | -24.218 +/- 2.261 |
| **Polar solvation Energy** | 206.400 +/- 29.228 | 118.488 +/- 34.637 | 207.832 +/- 45.946 | 173.854 +/- 39.115 |
| **Binding Energy** | -173.323 +/- 21.496 | -170.903 +/- 21.409 | -190.868 +/- 26.477 | -187.521 +/- 24.105 |

## Supporting information

**S1 File. Supporting information contains all the supporting tables and figures.**
(DOCX)

**S1 Graphical abstract.**
(TIF)

## Acknowledgments

We would like to thank all of the members of the Ben M'Sick Faculty of Science's as well as the physical chemistry of materials laboratory group and the heads of the chemistry department for their motivation and assistance in carrying out this work.

## Author Contributions

**Conceptualization:** Hatim Soufi, Said Belaaouad.

**Data curation:** Mouna Baassi, Hatim Soufi.

**Formal analysis:** Mouna Baassi, Mohamed Moussaoui.

**Investigation:** Mouna Baassi, Mohamed Moussaoui, Hatim Soufi, Sanchaita Rajkhowa.

**Methodology:** Mouna Baassi, Mohamed Moussaoui.

**Project administration:** Said Belaaouad.

**Supervision:** Sanchaita Rajkhowa, Ashwani Sharma, Said Belaaouad.

**Validation:** Mouna Baassi, Mohamed Moussaoui, Sanchaita Rajkhowa, Subrata Sinha.

**Visualization:** Mouna Baassi.

**Writing – original draft:** Mouna Baassi.

**Writing – review & editing:** Sanchaita Rajkhowa, Ashwani Sharma, Subrata Sinha, Said Belaaouad.

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
