## [Decision Letter · Decision Letter 0]

6 Jan 2023

PONE-D-22-30883Towards designing of a potential new HIV-1 protease inhibitor using QSAR study in combination with Molecular Docking, and Molecular Dynamics SimulationsPLOS ONE

Dear Dr. Rajkhowa,

Thank you for submitting your manuscript to PLOS ONE. After careful consideration, we feel that it has merit but does not fully meet PLOS ONE’s publication criteria as it currently stands. Therefore, we invite you to submit a revised version of the manuscript that addresses the points raised during the review process.

We look forward to receiving your revised manuscript.

Kind regards,

Arabinda Ghosh

Academic Editor

PLOS ONE

Journal Requirements:

3. Please ensure that you refer to Figure 11,12,13,14 and 15 in your text as, if accepted, production will need this reference to link the reader to the figure.

Reviewers' comments:

Reviewer's Responses to Questions

**Comments to the Author**

1. Is the manuscript technically sound, and do the data support the conclusions?

Reviewer #1: Partly

Reviewer #2: Partly

2. Has the statistical analysis been performed appropriately and rigorously? 

Reviewer #1: I Don't Know

Reviewer #2: Yes

3. Have the authors made all data underlying the findings in their manuscript fully available?

Reviewer #1: Yes

Reviewer #2: Yes

4. Is the manuscript presented in an intelligible fashion and written in standard English?

Reviewer #1: No

Reviewer #2: Yes

5. Review Comments to the Author

Reviewer #1: (i) The introduction can be more eloborate and informative focusing on drug resistance in HIV-1 protease.The problem statement can be improved to increase the readability of the manuscript. These papers used MD simulations to understand drug resistance in HIV-1 protease. They can also be cited to enrich the methodology of the manuscript:

(a) https://doi.org/10.1080/07391102.2021.2016486

(b) https://doi.org/10.1016/j.jmgm.2017.06.007

(c) https://doi.org/10.1080/07391102.2018.1492459

(d) http://dx.doi.org/10.1371/journal.pone.0087520

(ii) A flow chart of the work done may be included to appreciate the work.

(iii) Together, abstract and conclusions section should provide the message from the manuscript in one single reading.

(iv) There are also methodological issues in the paper. The authors have mentioned a multidrug resistant mutant structure PDB (3OUD) as WT and it is crystallized with a peptide mimic. The removal of this large inhibitor could be expected to introduce effects on the structure as it relaxes from the crystal form. The packing of water in the inhibitor binding groove could also be expected to have a substantial impact.This should be commented upon. Hence the authors need to choose carefully the WT and mutant structures.

(v) The authors should check what are the mutations present in multidrug resistant strain and how the synthesized compounds benifit in binding with these mutant strains than existing protease inhibitors (Darunavir). Whether these mutations are present in active site or non-active site or both the regions?

(vi) Each section of the "results and discussion" requires improvement. All the observations were

provided in detail. But what is the message from each sub section? -- This is not clear. How these

messages can be knitted to obtain the final conclusions drawn from the manuscript? -- This needs

to be efficiently done. Such an exercise shall improve the quality of the manuscript. This shall help

in deciding whether the conclusions are "trust worthy?"

(vii) The majority of the analyses presented by the authors are crude, and partially redundant- for example Rg and RMSD plots for example mirror one another closely and do not add new information. In hydrogen bond analysis, the authors have given an overall picture, but there are no details about the residues and ligand atoms that are involved in forming hydrogen bond.

(viii) There are many grammatical errors and typographical mistakes which needs to be corrected.

Reviewer #2: The comments are as follows:

The study is very interesting and has a good impact, although these major comments must be taken care of:

1. Authors must add Binding free energy calculations.

2. PCA & Free Energy Landscape analysis is a must-need with a comparison of DCCM.

3. Authors must take care of typo errors.

4. Radius of Gyration states the compactness of the protein-ligand complex but from (Figure 20 (B) the introduction of a new drug in blue colour is seen increasing from 57ns so how far has it got stabilized if the peaks are increasing?? Needed justification with a picture of Stepwise Trajectory analysis of 100ns run. [Refer to Figure 5 of this article: https://doi.org/10.1371/journal.pone.0263853]

5. In Figure 18.A of RMSF, in comparison to Lopinavir's RMSD curve's deviation the New Drug-MT's peaks are higher with higher fluctuation which signifies the incompatible binding with protein therefore how can authors conclude the comparison from Lopinavir that their proposed drug has a good RMSD??? Justifications needed.

6. PLOS authors have the option to publish the peer review history of their article (what does this mean?). If published, this will include your full peer review and any attached files.

Reviewer #1: **Yes: **C.S VASAVI

Reviewer #2: **Yes: **Nobendu Mukerjee; ORCID: https://orcid.org/0000-0002-7129-7003

---

## [Author Response · Author response to Decision Letter 0]

25 Feb 2023

Response to Reviewers

Reviewer 1:

We thank the reviewer for this thoughtful and thorough evaluation and trust; it has been priceless to enhancing the manuscript’s accuracy. We considered all the remarks and carefully addressed them one by one. Below are the responses to every query.

Reviewer’s Comment: The introduction can be more elaborate and informative focusing on drug resistance in HIV-1 protease. The problem statement can be improved to increase the readability of the manuscript. These papers used MD simulations to understand drug resistance in HIV-1 protease. They can also be cited to enrich the methodology of the manuscript:

(a) https://doi.org/10.1080/07391102.2021.2016486

(b) https://doi.org/10.1016/j.jmgm.2017.06.007

(c) https://doi.org/10.1080/07391102.2018.1492459

(d) http://dx.doi.org/10.1371/journal.pone.0087520

Our Response: We thank again the esteemed reviewer for giving their valuable comments. As suggested by the reviewer, we did elaborate the introduction making it more informative focusing on drug resistance in HIV-1 protease along with the other sections of our paper. In addition, the proposed references were added in the manuscript to support the information included. (in Introduction section, page no. 2)

Introduction

Human Immunodeficiency Virus (HIV) is one of the most challenging viruses in medicine, causing severe complications related to human health [1]. HIV which is responsible for Acquired Immunodeficiency Syndrome (AIDS), still has no cure for more than three decades [2]. This is the main reason why synthesized drugs have been used in combinations to treat HIV infection [3], [4]. Highly active antiretroviral therapy (HAART) attacks multiple stages of the HIV viral life cycle and stops the virus from making copies of itself in the body thus leading to a reduction in mortality and morbidity rates of HIV/AIDS [3], [5]–[7].

Antiretroviral therapy plays an essential role in the treatment of HIV/AIDS, but the accelerated evolution of multidrug-resistant (MDR) strains of HIV-1 protease (PR) and poor oral bioavailability and side effects have firmly restricted long-term treatment benefits [8], [9]. 

PIs are supposed to overcome the replication of viruses. However, some residual viral activity endures throughout the therapy process, leading to the development of drug-resistant strains with various mutations that decrease protease affinity for the inhibitors. The mutations are detected not precisely inside the active site where they directly affect the inhibitor binding but also outside the binding site [10]–[12].

Corresponding to the International AIDS Society, 23 mutations in 16 codons of the protease gene relevant to significant drug resistance to PIs were highlighted using phenotypic resistance assays [13].

Therefore, the design of new HIV-1 PIs has become an obligation. In order to discover new drugs, looking forward to amplifying the inhibitory activity and according to the strategy to defeat drug resistance, a series of 33 compounds were synthesized and evaluated in previous work for their antiretroviral activities [14]. The primary purpose of this study is to develop a Quantitative Structure Activity Relationship (QSAR) model able to relate the structural features (descriptors) to the biological activity of these drug candidates against HIV-1 protease. 

The QSAR method is based on computational methods, aiming at relating the activity (y) to the chemical properties (x), y = f(x) [15]. To achieve this, we need a series of compounds with well-known biological activities (y), and for each compound, we compute several descriptors (x) using various software, incorporating the DFT method [16],[17].

Once the QSAR model is elaborated and statistically validated, it can be used for the prediction, analysis, and estimation of new elements with convenient activities, minimizing time, effort, and charges [18].

Reviewer’s Comment: A flow chart of the work done may be included to appreciate the work.

Our Response: As suggested by the reviewer, a flow chart was included, and placed in the introduction section of the paper. (in Introduction section, page no. 2, line no. 30-31 + Fig 1)

Introduction

The flow chart mentioned above (Fig 1) covers an overview of the multiple axes elaborated along with this research.

Fig 1: Flow chart of the current work

Reviewer’s Comment: Together, abstract and conclusions section should provide the message from the manuscript in one single reading.

Our Response: As suggested by the reviewer, significant modifications were incorporated in the abstract and the conclusion sections of the manuscript to provide the message from the manuscript in one single reading. (in Abstract section, page no. 1, line no. 1-5, 14-21 and Conclusion section, page no. 24-25, line no 10-16 )

Abstract

Human Immunodeficiency Virus type 1 protease (HIV-1 PR) is one of the most challenging targets of antiretroviral therapy used in the treatment of AIDS-infected people. The performance of protease-inhibitors (PIs) is limited by the development of protease mutations that can promote resistance to the treatment. The current study was carried out using statistics and bioinformatics tools.

A series of thirty-three compounds with known enzymatic inhibitory activities against HIV-1 protease was used in this paper to build a mathematical model relating the structure to the biological activity. These compounds were designed by software; their descriptors were computed using various tools, such as Gaussian, Chem3D, ChemSketch and MarvinSketch. Computational methods generated the best model based on its statistical parameters. The model's applicability domain (AD) was elaborated. Furthermore, one compound has been proposed as efficient against HIV-1 protease with comparable biological activity to the existing ones; this drug candidate was evaluated using ADMET properties and Lipinski's rule. Molecular Docking performed on Wild Type, and Mutant Type HIV-1 proteases allowed the investigation of the interaction types displayed between the proteases and the ligands, Darunavir (DRV) and the new drug (ND). Molecular dynamics simulation was also used in order to investigate the complexes' stability allowing a comparative study on the performance of both ligands (DRV & ND). Our study suggested that the new molecule showed comparable results to that of darunavir and maybe used for further experimental studies. Our study may also be used as pipeline to search and design new potential inhibitors of HIV-1 proteases.

Conclusion

Various softwares have been used in this study in order to generate a reliable model relating the biological activity of new HIV-1 protease inhibitors to their physicochemical parameters. The generated model showed a high predictability efficiency regarding its statistical parameters. The applicability domain was also generated to frame the workspace (only compounds with features with greater similarity to those included in the training set can be used). Regarding the proposed model, the biological activity of the new HIV-1 protease inhibitors can be increased by increasing the three variables' values; the Energy Gap (EGap); the Polar Surface Area (PSA) and the Dreiding Energy (DE) (positively related to the activity), and decreasing the Henry's Law Constant value (negatively related to the activity). A new drug was proposed based on the model generated with a biological activity higher than the known drug compounds' activities. Afterwards, the molecular docking study was performed on the wild-type and the mutant-type HIV-1 proteases to predict the best conformation displayed by two ligands, the New Drug and Darunavir as an approved FDA drug. Moreover, molecular dynamics simulation was performed to study the stability of the complexes (WT-DRV, WT-ND, MT-DRV & MT-ND); results disclosed some interesting results related to the new drug, therefore, the new drug may be considered as a potential inhibitor against multi-drug-resistant (MDR) strains of HIV-1 protease (PR) and may be tested experimentally. 

Reviewer’s Comment: There are also methodological issues in the paper. The authors have mentioned a multidrug resistant mutant structure PDB (3OUD) as WT and it is crystallized with a peptide mimic. The removal of this large inhibitor could be expected to introduce effects on the structure as it relaxes from the crystal form. The packing of water in the inhibitor binding groove could also be expected to have a substantial impact. This should be commented upon. Hence the authors need to choose carefully the WT and mutant structures.

Our Response: We thank the reviewer once again for the valuable time he has given in evaluating this manuscript as well as for his valuable remarks that helped us to enhance the accuracy of our results. As the reviewer suggested, we did chose carefully the proteases structures from Protein Data Bank library. (in Molecular Docking section, page no. 14, line no. 4-7, 11-12)

Molecular Docking 

Molecular docking study was carried out with the aim of predicting the best conformation of the HIV-1 protease of both types (mutant and wild), on the one hand; combined to the proposed compound as a new efficient drug candidate (ND), on the other hand; combined to an FDA approved drug called Darunavir (DRV). We selected both types of the HIV-1 protease (WT and MT) as receptors. The structures of the wild type (WT) as well as the mutant type (MT) proteases were downloaded from Protein Data Bank (PDB), their PDP codes are respectively: (4LL3-Structure of wild-type HIV-1 protease in complex with Darunavir) (Fig 5) and (3TTP-Structure of multiresistant HIV-1 protease in complex with Darunavir) (Fig 6). Their original ligands were eliminated using Discovery Studio, polar hydrogens were added and the proteins were saved in PDB format, and then saved in PDBQT format using Autodock MGL Tools. The ligand proposed as a new efficient drug was earlier designed and optimized using Gaussian, then saved in PDBQT format by Autodock MGL tools (Fig 7); in addition, DRV was taken from the crystal structures downloaded from Protein Data Bank (Fig 8).

Reviewer’s Comment: The authors should check what are the mutations present in multidrug resistant strain and how the synthesized compounds benefit in binding with these mutant strains than existing protease inhibitors (Darunavir). Whether these mutations are present in active site or non-active site or both the regions?

Our Response: We do thank the reviewer for all their priceless comments including this one. As requested by the esteemed reviewer, we did investigate the mutations present in the structure of the multi-drug resistant in HIV-1 protease and we highlighted the benefit in binding with the new drug as compared to Darunavir. (in Supplementary Information file, page no. 7-8)

Mutations occurring in the mutant type protease 3TTP

Multiple mutations are present in the multidrug resistant 3TTP as compared to the wild type, notably V13, R20, I32, F33, D35, I36, K41, T43, V47, M54, V62, V63, V71, T72, S73, P74, L82, V89 and L93 (S6 Table). 

S6 Table: The sequence of the wild type protease and its alignment with the mutant type sequence. The major DRV resistance-associated mutations are mentioned in green, minor mutations in blue and other mutations are colored in red [1]

WT PQITLWQRPLVT IKIGGQLKEALLDTGADDTVLEEMNLPGRWKPKMI GGIGGF I KVRQYDQI L IEICGHKAIGTVLVGPTPVNIIGRNLLTQ IGCTLNF

MT PQITLWQRPLVTVKIGGQLREALLDTGADDTI FED I NLPGKWTPKMVGGIGGFMKVRQYDQVVIEICGHK VTSPVLVGPTPLNIIGRNVLTQLGCTLNF

In the one hand, as shown in the table below (S7 Table), there is a major DRV resistance-associated mutation Val47 included in the active site of 3TTP while docked to Darunavir (S1 Fig), in the other hand, the active site of 3TTP while docked to the new drug is free of any possible mutation (S2 Fig), supporting the previous results (docking study) disclosing the affinity values of MT-DRV and MT-ND; -9.9 and -10.4 Kcal/mol respectively, we can conclude that effectively the complex compound MT-ND is showing higher stability as compared to MT-DRV. 

S7 Table: The complex compounds’ (MT-DRV & MT-ND) active site residues

MT-DRV Chaine A: Asp25, Asp30, Ile50, Pro81 and Ile84. 

Chaine B: Ala28, Ile32, Val47, Gly48, Ile50 and Ile84. 

 MT-ND Chaine A: Arg8, Gly27, Ala28, Gly49, Ile50, Pro81 and Ile84. 

Chaine B: Asp25, Ala28, Ile50 and Ile84. 

 

S1 Figure: 3D-Structure of MT-DRV, mutations are colored in blue, the active site residues in purple and the mutation Val47 in yellow (a residue present in the active site).

S2 Figure: 3D-Structure of MT-ND, mutations are colored in blue and the active site residues in purple

Reviewer’s Comment: Each section of the "results and discussion" requires improvement. All the observations were provided in detail. But what is the message from each sub section? 

-- This is not clear. How these messages can be knitted to obtain the final conclusions drawn from the manuscript? 

-- This needs to be efficiently done. Such an exercise shall improve the quality of the manuscript. This shall help in deciding whether the conclusions are "trust worthy?"

Our Response: As suggested by the reviewer, required modifications were elaborated in the results and discussion section, therefore all interpretation are written briefly and efficiently to support all our findings. (in Molecular Docking section, pages no. 15,17 and Molecular Dynamics Simulation section, pages no. 17-22)

Reviewer’s Comment: The majority of the analyses presented by the authors are crude, and partially redundant- for example Rg and RMSD plots for example mirror one another closely and do not add new information. In hydrogen bond analysis, the authors have given an overall picture, but there are no details about the residues and ligand atoms that are involved in forming hydrogen bond.

Our Response: Once again, we would like to thank the reviewer for his highly appreciated remarks as they enhance the accuracy of our research. As mentioned above, major modifications were done in accordance with the main purpose of the research to support the results. (in Molecular Docking section, pages no. 15,17 and Molecular Dynamics Simulation section, pages no. 17-22)

Molecular Docking

Command prompt and Vina folder were used in order to run the Docking. Different conformations of the ligand binding modes for both types were obtained with their respective binding energies (kcal/mol) after the accomplishment of the docking runs; the best pose is the one with the lowest affinity value.

The best-ranked poses based on their binding affinities are selected for farther analysis; figures (Fig 9, 10, 13 & 14) represent the 2D-binding interactions in the active site of the proteases; wild type and mutant type with Darunavir and the new drug. Figures (Fig 11, 12, 15 & 16) disclose the 3D-interactions for the same compounds (WT-ND, MT-ND, WT-DRV & MT-DRV).  

The interactions between the ligands (ND & DRV) and the proteases were visualized using Discovery Studio (Table 7). Active residues interacting with the ligands (ND & DRV) are also disclosed (S5 Table). Moreover, atoms from ligands and residues interacting with each other to form hydrogen bonds are mentioned (S8 Table).

Table 7: Complex compounds' affinity values and the number of different interaction types. 

Protein Ligand Nbre of Hydrogen bonds Nbre of hydrophobic interactions VDW Binding affinity

WT DRV 4 10 16 -9.6

 ND 4 12 15 -10.2

MT DRV 4 11 21 -9.9

 ND 5 7 17 -10.4

Based on the Molecular Docking analysis; results lead us to conclude that the complex compounds (WT-ND & MT-ND) with binding affinity values of -10.2 kcal/mol & -10.4 kcal/mol respectively, display a higher stability as compared to (WT-DRV & MT-DRV). 

 

Molecular dynamics simulation

To evaluate the native proteins' stability (WT & MT), as well as the docked compounds' (WT-DRV, WT-ND, MT-DRV & MT-ND), a computational process is carried out through the Molecular Dynamics simulation (MD) study, allowing structural analysis at the atomic level, aiming at investigating the motion of the four complex compounds and the native proteins.

Therefore, MD simulations were administered in nine plots, with 100ns for each, using the best poses generated based on the docking study performed previously, the compounds were carried out in water simulations separately. Further, the stability analysis was performed through several techniques, namely: Root Mean Square Deviation (RMSD), Root Mean Square fluctuation (RMSF) and the Radius of Gyration (Rg). 

Root Means Square Deviation (RMSD)

RMSD stands for Root Means Square Deviation, it is a numerical measurement, it estimates the approximate distance between a band of atoms, mainly, backbone atoms of a protein plotted against time. The Root Means Square Deviation value is typically a measure of how much the protein's structure has been modified over time in comparison to the starting point. Further, if the RMSD of the protein presents considerable fluctuations, then no equilibrium is reached, therefore, more simulation time is required for better results.

As the RMSD plots display (Fig 17), the native proteins (WT and MT) do not show any promising stability within the simulation time especially for the wild type protease. Regarding the RMSD plots (Fig 17 (B)) for the two complexes (WT-DRV and WT-ND), it is highly clear that these compounds are showing lower fluctuations than the native protein (WT) within the simulation time. As for the complexes (MT-ND and MT-DRV), they're showing as well lower fluctuations as compared to the native protein (MT) within the simulation time (Fig 17 (A)). However, WT-ND and MT-ND complexes are showing promising results comparable to those of Darunavir in terms of fluctuations. 

Fig 17: The root means square deviation (RMSD) plots of MT (A) and WT (B) proteases virgin as with ligands (ND and DRV) during 100ns of molecular dynamics simulation.

Root Means Square Fluctuation (RMSF)

The Root Mean Square fluctuation (RMSF) measures the approximate deviation of a particle over time from a reference position at a specific temperature and pressure. The RMSF analysis illuminates the fluctuations of residues during the MD simulation time. 

Considering the graphics, for the wild type and the mutant type proteases (Fig 18 (A), (B), (C) & (D)), both chains (A & B) are displaying slightly similar fluctuations in some regions, and highly non-similar fluctuations in the other regions, leading us to conclude that for all complexes (WT-DRV, WT-ND, MT-DRV & MT-ND) regardless the chain, the new drug and Darunavir are significantly influencing the fluctuations of the proteins' residues in most regions. 

Fig 18: The root means square fluctuation (RMSF) plots of MT (chain (A) (A) and chain (B) (B)) and WT (chain (A) (C) and chain (B) (D)) proteases without and with ligands (ND and DRV) during the period of simulation.

Radius of gyration (Rg)

The radius of gyration is an interesting parameter as well to investigate the motion of a protein as well as its stability; it describes the compactness of the protein during the simulation time.

For the Mutant Type protease (Fig 19 (A)), the radius of gyration of the complex compound MT-ND is higher in value as compared to the MT native protein and the complex compound in presence of DRV, causing eventually higher flexibility of the compound MT-ND. For the Wild Type protease (Fig 19 (B)), the plots show that the complex compound WT-ND reveals more compactness with lower radius of gyration values as compared to the complex compound WT-DRV and the WT native protein within the simulation time, inducing less flexibility, which means higher potential of stability for the complex WT-ND. 

Fig 19: Graphs representing the Radius of gyration (Rg) values for MT (A) and WT (B) proteases virgin as with ligands (ND and DRV) during the period of simulation.

Hydrogen Bonds

Hydrogen bonds are primordial in drug specificity and stability, so the determination of H-bond number in complex compounds is essential to check its contribution to the overall stability of each system and further conduct a comparative study including all complex compounds in question. 

The figure (Fig 20) shows that during the MD simulation period (100ns), the complex MT-ND's graph is showing up to seven hydrogen bonds by the end of the simulation time, while the MT-DRV complex compound's graph is showing a few hydrogen bonds during the first 40ns as compared to MT-ND, then significantly increasing at 60ns displaying ten hydrogen bonds then decreasing to seven by the end of the simulation time (Fig 20 (A)). In contrast, for the complex compound WT-DRV, the number of hydrogen bonds is consistently decreasing from 8 to 5 while the compound WT-ND displays up to five hydrogen bonds with no significant decrease compared to the WT-DRV compound during the simulation time (Fig 20 (B)).

We can conclude that whether the wild type or the mutant type proteases, when docked to the new drug, the number of hydrogen bonds is likely to be the same with no significant change as compared to the complex compounds with Darunavir that shows a decreasing number of hydrogen bonds during the simulation time. 

Fig 20: Graphs showing the number of hydrogen bonds (at every 20 ns) along with the simulation time for complex compounds containing MT (A) and WT (B) proteases

Hydrophobic interactions

Hydrophobic interactions are non-bonded interactions between the protein and the ligand, which play a major role in the stability of complexes.

As shown below, considering the wild type protease (Fig 21 (B), both complexes WT-DRV and WT-ND show highly similar numbers of hydrophobic interactions during the simulation time. In contrast, for the mutant type protease (Fig 21 (A), the complexes MT-DRV and MT-ND, the number of hydrophobic interactions for the complex compound MT-DRV is significantly higher than the number of hydrophobic interactions for the complex compound with the new drug MT-ND.

We can conclude that for the wild-type protease, the new drug significantly competes with Darunavir, displaying similar numbers of hydrophobic interactions at every 20 ns of the simulation time. However, Darunavir is showing highly promising results for the mutant-type protease compared to the new drug in terms of hydrophobic interactions. 

Fig 21: Graphs showing the number of hydrophobic interactions (at every 20 ns) along with the simulation time for complex compounds containing MT (A) and WT (B) proteases

Solvent Accessible Surface Area (SASA) 

The accessible surface area (ASA) or solvent-accessible surface area (SASA) is the surface area of a biomolecule that is accessible to a solvent. 

Based on the graphics (Fig 22), the new drug, when combined to the wild type protease, is showing promising results regarding the significant decrease of the ASA values since 40ns to the end of the simulation time (Fig 22 (B)), but for the mutant type, the ASA values are not promising on the ground that the graphics are displaying increasing values starting from 60ns of the simulation time (Fig 22 (A)). 

We can conclude that the new drug is comparable to Darunavir during the last 30ns of the simulation time for the wild type protease while no possible competition is investigated for the mutant type on the ground that the graphic is showing significant ASA values for the complex MT-ND as compared the MT-DRV mainly during the last 40ns of the simulation time. 

Fig 22: Time evolution (at every 20 ns) of solvent accessible surface area (SASA) or of the wild type and the mutant type proteases along with DRV and ND

Reviewer’s Comment: There are many grammatical errors and typographical mistakes which needs to be corrected.

Our Response: As mentioned by the esteemed reviewer, the errors were corrected one by one.

Reviewer 2:

We thank the reviewer for considering and evaluating the manuscript, it has been mandatory for enhancing the manuscript accuracy. We have gone through all the comments and remarks and thoroughly addressed them one by one. Underneath are the responses to every problem and question.

Reviewer’s Comment: Authors must add Binding free energy calculations.

Our Response: As suggested by the esteemed reviewer, the binding free energy and its corresponding component obtained from the MM-PBSA calculations were listed in a table. (in Binding Free Energy Calculation section, pages no. 23-24)

Binding Free Energy Calculation

Molecular dynamics simulations were used to calculate binding free energy using the MM-PBSA method. Snapshots were extracted at every 1 ns of stable intervals from 70-100 ns MD trajectory. The binding free energy and its corresponding component obtained from the MM-PBSA calculations are listed (Table 8). 

Table 8: Average MM-PBSA free energies for proteins (WT and MT) and ligands (DRV and ND)

Energy WT-Darunavir

(kJ/mol) WT- New Drug

(kJ/mol) MT- Darunavir

(kJ/mol) MT- New Drug

(kJ/mol)

van der Waal -225.893

+/- 16.265 -219.745

+/- 17.427 -250.856

+/- 22.852 -248.746

+/- 26.755

Electrostatic -119.917

+/- 24.965 -75.793

+/- 20.319 -110.692

+/- 39.922 -98.412

+/- 26.172

SASA Energy -21.913

+/- 1.188 -22.853

+/- 1.809 -22.152

+/- 1.964 -24.218

+/- 2.261

Polar solvation

Energy 206.400

+/- 29.228 118.488

+/- 34.637 207.832

+/- 45.946 173.854

+/- 39.115

Binding Energy -173.323

+/- 21.496 -170.903

+/- 21.409 -190.868

+/- 26.477 -187.521

+/- 24.105

The results indicate that for both wild and mutant type protease, Darunavir is showing a binding affinity of -173.323 kJ/mol and -190.868 kJ/mol, respectively, which is slightly higher than the New Drug (-170.903 kJ/mol and -187.521 kJ/mol, respectively). 

van der Waals, Electrostatic and SASA energy played a crucial role in binding energy and complex stability. In contrast, polar solvation energy has an opposite effect causing binding energy to depend on its unfavorable positive value. Among different energy terms, the contribution of van der Waals energy towards total binding energy is superior.

Compilation of the data demonstrated that although the binding of Darunavir to both wild and mutant HIV protease is better, the binding of the new drug is comparable to that of Darunavir in both wild and wild mutant type. This is illustrated by the different analyses that have been used so far. Thus, the new drug may also be considered a potential inhibitor against multi-drug resistant HIV and may be tested experimentally. 

Reviewer’s Comment: PCA & Free Energy Landscape analysis is a must-need with a comparison of DCCM.

Our Response: We thank the reviewer once again for the valuable comments. Regarding the PCA analysis, Free Energy Landscape analysis and DCCM comparison, we believe that these analyses will not be of much value to this work. This study is not concerned with modeling of the wild or the mutant type proteins. The crystal structures of both the wild as well as the mutant proteins have been downloaded from the RCSB PDB.

Principal component analysis (PCA) is a covariance-matrix-based mathematical technique and is used to reduce a multidimensional complex set of variables to a lower dimension along which the diffusive properties at all stages of protein folding can be identified. However, we are only studying the stability of the ligands onto the proteins. Therefore, the PCA will not provide any extra insight in this study. 

Similarly, the energy landscape theory of protein folding is a statistical description of a protein's potential surface. It assumes that folding occurs through organizing an ensemble of structures rather than through only a few uniquely defined structural intermediates. As this study does not concern the protein folding/unfolding, this analysis will also not provide much additional insight.

Additionally, DCCMs are usually utilized to characterize correlated motions between residues of receptors. In our study, almost all the MD analyses suggests that both the ligands are in fact stable within the binding site. Therefore, the suggested analysis is not included in this manuscript.

We thank the reviewer once again for the suggestion and would definitely try to include such analyses in our future study.

Reviewer’s Comment: Authors must take care of typo errors.

Our Response: As mentioned by the esteemed reviewer, the grammatical and typo errors were corrected one by one.

Reviewer’s Comment: Radius of Gyration states the compactness of the protein-ligand complex but from (Figure 20 (B) the introduction of a new drug in blue colour is seen increasing from 57ns so how far has it got stabilized if the peaks are increasing?? Needed justification with a picture of Stepwise Trajectory analysis of 100ns run. [Refer to Figure 5 of this article: https://doi.org/10.1371/journal.pone.0263853]

Our Response: We once again thank the esteemed reviewer for the suggestions. However as suggested by reviewer 1, the structures of both the wild and mutant proteins have been changed. Therefore, the results and analyses have also been modified. (in Radius of gyration section, pages no. 20) 

Radius of gyration (Rg)

The radius of gyration is an interesting parameter as well to investigate the motion of a protein as well as its stability; it describes the compactness of the protein during the simulation time.

For the Mutant Type protease (Fig 19 (A)), the radius of gyration of the complex compound MT-ND is higher in value as compared to the MT native protein and the complex compound in presence of DRV, causing eventually higher flexibility of the compound MT-ND. For the Wild Type protease (Fig 19 (B)), the plots show that the complex compound WT-ND reveals more compactness with lower radius of gyration values as compared to the complex compound WT-DRV and the WT native protein within the simulation time, inducing less flexibility, which means higher potential of stability for the complex WT-ND. 

Fig 19: Graphs representing the Radius of gyration (Rg) values for MT (A) and WT (B) proteases virgin as with ligands (ND and DRV) during the period of simulation.

Reviewer’s Comment: In Figure 18.A of RMSF, in comparison to Lopinavir's RMSD curve's deviation the New Drug-MT's peaks are higher with higher fluctuation which signifies the incompatible binding with protein therefore how can authors conclude the comparison from Lopinavir that their proposed drug has a good RMSD??? Justifications needed.

Our Response: We once again thank the esteemed reviewer for the suggestions. However as suggested by reviewer 1, the structures of both the wild and mutant proteins have been changed. Therefore, the results and analyses have also been modified. (in Root Means Square Deviation (RMSD) section, pages no. 18 and in Root Means Square Fluctuations (RMSF) section, pages no. 19)

Root Means Square Deviation (RMSD)

RMSD stands for Root Means Square Deviation, it is a numerical measurement, it estimates the approximate distance between a band of atoms, mainly, backbone atoms of a protein plotted against time. The Root Means Square Deviation value is typically a measure of how much the protein's structure has been modified over time in comparison to the starting point. Further, if the RMSD of the protein presents considerable fluctuations, then no equilibrium is reached, therefore, more simulation time is required for better results.

As the RMSD plots display (Fig 17), the native proteins (WT and MT) do not show any promising stability within the simulation time especially for the wild type protease. Regarding the RMSD plots (Fig 17 (B)) for the two complexes (WT-DRV and WT-ND), it is highly clear that these compounds are showing lower fluctuations than the native protein (WT) within the simulation time. As for the complexes (MT-ND and MT-DRV), they're showing as well lower fluctuations as compared to the native protein (MT) within the simulation time (Fig 17 (A)). However, WT-ND and MT-ND complexes are showing promising results comparable to those of Darunavir in terms of fluctuations. 

Fig 17: The root means square deviation (RMSD) plots of MT (A) and WT (B) proteases virgin as with ligands (ND and DRV) during 100ns of molecular dynamics simulation.

Root Means Square Fluctuation (RMSF)

The Root Mean Square fluctuation (RMSF) measures the approximate deviation of a particle over time from a reference position at a specific temperature and pressure. The RMSF analysis illuminates the fluctuations of residues during the MD simulation time. 

Considering the graphics, for the wild type and the mutant type proteases (Fig 18 (A), (B), (C) & (D)), both chains (A & B) are displaying slightly similar fluctuations in some regions, and highly non-similar fluctuations in the other regions, leading us to conclude that for all complexes (WT-DRV, WT-ND, MT-DRV & MT-ND) regardless the chain, the new drug and Darunavir are significantly influencing the fluctuations of the proteins' residues in most regions. 

Fig 18: The root means square fluctuation (RMSF) plots of MT (chain (A) (A) and chain (B) (B)) and WT (chain (A) (C) and chain (B) (D)) proteases without and with ligands (ND and DRV) during the period of simulation.

---

## [Decision Letter · Decision Letter 1]

3 Apr 2023

Towards designing of a potential new HIV-1 protease inhibitor using QSAR study in combination with Molecular Docking, and Molecular Dynamics Simulations

PONE-D-22-30883R1

Dear Dr. Sanchaita Rajkhowa

We’re pleased to inform you that your manuscript has been judged scientifically suitable for publication and will be formally accepted for publication once it meets all outstanding technical requirements.

Kind regards,

Arabinda Ghosh

Academic Editor

PLOS ONE

Reviewers' comments:

Reviewer's Responses to Questions

**Comments to the Author**

1. If the authors have adequately addressed your comments raised in a previous round of review and you feel that this manuscript is now acceptable for publication, you may indicate that here to bypass the “Comments to the Author” section, enter your conflict of interest statement in the “Confidential to Editor” section, and submit your "Accept" recommendation.

Reviewer #1: All comments have been addressed

Reviewer #2: All comments have been addressed

2. Is the manuscript technically sound, and do the data support the conclusions?

Reviewer #1: Partly

Reviewer #2: Yes

3. Has the statistical analysis been performed appropriately and rigorously? 

Reviewer #1: I Don't Know

Reviewer #2: Yes

4. Have the authors made all data underlying the findings in their manuscript fully available?

Reviewer #1: Yes

Reviewer #2: Yes

5. Is the manuscript presented in an intelligible fashion and written in standard English?

Reviewer #1: Yes

Reviewer #2: Yes

6. Review Comments to the Author

Reviewer #1: Kindly check the units for the binding affinity, as it is reported in KJ/mol. Shouldn't it be kcal/mol?

Reviewer #2: All the comments have been addressed, now the manuscript seems good and acceptable for this journal.

7. PLOS authors have the option to publish the peer review history of their article (what does this mean?). If published, this will include your full peer review and any attached files.

Reviewer #1: No

Reviewer #2: No

---

## [Editor Report · Acceptance letter]

11 Apr 2023

PONE-D-22-30883R1 

Towards designing of a potential new HIV-1 protease inhibitor using QSAR study in combination with Molecular Docking and Molecular Dynamics Simulations 

Dear Dr. Rajkhowa:

I'm pleased to inform you that your manuscript has been deemed suitable for publication in PLOS ONE. Congratulations! Your manuscript is now with our production department. 

Kind regards, 

on behalf of

Dr. Arabinda Ghosh 

Academic Editor

PLOS ONE